# Forethought and Hindsight in Credit Assignment

**Veronica Chelu**
Mila, McGill University

**Doina Precup**
Mila, McGill University, DeepMind

**Hado van Hasselt**
DeepMind

## Abstract

We address the problem of credit assignment in reinforcement learning and explore fundamental questions regarding the way in which an agent can best use additional computation to propagate new information, by planning with internal models of the world to improve its predictions. Particularly, we work to understand the gains and peculiarities of planning employed as forethought via forward models or as hindsight operating with backward models. We establish the relative merits, limitations and complementary properties of both planning mechanisms in carefully constructed scenarios. Further, we investigate the best use of models in planning, primarily focusing on the selection of states in which predictions should be (re)-evaluated. Lastly, we discuss the issue of model estimation and highlight a spectrum of methods that stretch from explicit environment-dynamics predictors to more abstract planner-aware models.

## 1 Introduction

Credit assignment, i.e. determining how to correctly associate delayed rewards with states or state-action pairs, is a crucial problem for reinforcement learning (RL) agents (Sutton and Barto, 2018). Model-based RL (MBRL) agents gradually learn a model of the rewards and transition dynamics through interaction with their environments and use the estimated model to find better policies or predictions (e.g., Sutton, 1990a; Peng and Williams, 1993; Moore and Atkeson, 1993; McMahan and Gordon, 2005; Sutton et al., 2012; Farahmand et al., 2017; Farahmand, 2018; Abachi et al., 2020; Wan et al., 2019; Silver et al., 2017; Schrittwieser et al., 2019; Deisenroth et al., 2015; Hester and Stone, 2013; Ha and Schmidhuber, 2018; Oh et al., 2017). However, the efficiency of MBRL depends on learning a useful model for its purpose. In this paper, we focus specifically on the use of models for value credit assignment.

Broadly, we refer to *planning* as any internal processing that an agent can perform without additional experience to improve prediction and/or performance. Within this nomenclature, we define *models* as knowledge about the internal workings of the environment, which can be routinely re-used through planning. One way of using models is by *forethought* or *trying things in your head* (Sutton and Barto, 1981), which requires learning to predict aspects of the future and planning *forward*, or *in anticipation* to achieve goals. Dyna-style planning (Sutton, 1990a) chooses a (possibly hypothetical) state and action and predicts the resulting reward and next state, which are then used for credit assignment.

The origins of the chosen state and action are referred to as *search control strategies*. Lin (1992) proposed to use states actually experienced, and introduced the idea of replaying prior experience (Lin, 1992; Mnih et al., 2015). Combinations of these two approaches result in *prioritized sweeping* variants (Moore and Atkeson, 1993; Peng and Williams, 1993; McMahan and Gordon, 2005), which generalize the idea of replaying experience in backward order (Lin, 1992) by prioritizing states based on the potential improvement in the value function estimate upon re-evaluation. From high priority states, forward models are used to perform additional value function updates, increasing computational efficiency (e.g., van Seijen and Sutton, 2013). An investigation into *source-control strategies* (Pan et al., 2018) reveals the utility of additional prioritization for guiding learning towards relevant, causal or interesting states.

In this paper, we work to understand how different phenomena caused by the structure of an environment favor the use of forward or backward planning mechanisms for credit assignment. We define *backward models* as learning to predict potential predecessors of observed states, either through explicit predictors of the environment or via planner-aware models, where the latter account for how the planner performs credit assignment. Backward models are interesting from two standpoints: (i) they can be used to causally change predictions or behaviour in hindsight, thereby naturally prioritizing states where predictions need to be (re)-evaluated; (ii) modeling errors in backward models can sometimes be less detrimental, because updating misspecified imaginary states with real experience may be less problematic as the reverse (van Hasselt et al., 2019; Jafferjee et al., 2020). We hope that additional understanding of the mechanisms of backward planning paves the way for new, principled algorithms that use models to seamlessly integrate both forethought and hindsight (as had been the case in traditional planning methods – LaValle, 2006).

The estimation and usage of models can be done in many ways (van Hasselt et al., 2019; Van Seijen and Sutton, 2015; Parr et al., 2008; Wan et al., 2019). The conventional approach is to learn explicit predictors of the environment which, if accurate enough, lead to good policies. However, no model is perfect. Model error is dependent on the choice of predictors and whether the true environment dynamics can be represented. *Planner-aware model learning* suggests learning instead only aspects of the environment relevant to the way in which the model is going to be used by the planner. This class of methods (Farahmand et al., 2017; Farahmand, 2018; Joseph et al., 2013; Silver et al., 2017; Oh et al., 2017; Farquhar et al., 2017; Luo et al., 2019; D'Oro et al., 2019; Schrittwieser et al., 2019; Abachi et al., 2020; Ayoub et al., 2020) incorporates knowledge about the value function or policy when learning the model. We describe a spectrum of methods for model estimation. On one end, we have environment predictors that rely on maximum likelihood estimation based on supervised learning. Towards the opposite end, constraints can be progressively relaxed by accounting how planners use the models, ultimately leading to fully abstract models – i.e. learnable black boxes (Xu et al., 2020; Oh et al., 2020).

**Contributions**  We investigate the emergent properties of planning forward and backward with learned models of the world. We justify the use of backward models in identifying relevant states from which to recompute prediction errors, for which we establish available design choices to be made with respect to what the model represents, how it is estimated, and how it is parametrized. We review these in the broader context of model estimation. Finally, we conduct an empirical study on illustrative prediction and control tasks, which builds intuition and provides evidence for our findings.

## 2 Background and Notation

We consider the usual RL setting (Sutton and Barto, 2018) in which an agent interacts with an environment, modelled as a (discounted) *Markov decision process* (MDP) (Puterman, 1994) $(\mathcal{S}, \mathcal{A}, \mathcal{P}^\star, r^\star, \gamma)$, with state space $\mathcal{S}$, action space $\mathcal{A}$, state-transition distribution $\mathcal{P}^\star : \mathcal{S} \times \mathcal{A} \to \mathcal{P}(\mathcal{S})$ (where $\mathcal{P}(\mathcal{S})$ is the set of probability distributions on the state space and $\mathcal{P}^\star(s'|s, a)$ denotes the probability of transitioning to state $s'$ from $s$ by choosing action $a$), reward function $r^\star : \mathcal{S} \times \mathcal{A} \to \mathbb{R}$, and discount $\gamma \in (0, 1)$. A policy $\pi$ maps states to distributions over actions; $\pi(a|s)$ denotes the probability of choosing action $a$ in state $s$. A *Markov reward process* (MRP) is specified as $(\mathcal{S}, \mathcal{P}^\star, r^\star, \gamma)$. The return at time $t$ is defined as: $G_t = \sum_{k=0}^\infty \gamma^k R_{t+k+1}$. The value function maps each state $s \in \mathcal{S}$ to the expected value of the return: $v_\pi(s) = \mathbb{E}[G_t|S_t = s, A_n \sim \pi(\cdot|S_n), \forall n \geq t]$. Value-based methods can be used for control by approximating the optimal action-value function $q^\star(s, a)$, representing the expected return when following the optimal policy, conditioned on a state-action pair.

In general, the agent does not directly observe the true state of the environment $s$, and instead observes or constructs a feature vector $\mathbf{x}(s)$. The value function can then be approximated using a parametrized function $v_\mathbf{w}(s) \approx v_\pi(s)$ with $\mathbf{w} \in \mathbb{R}^d$ and $d$ the size of feature representation.

This estimate can be linear: $v_\mathbf{w}(s) = \mathbf{w}^\top \mathbf{x}(s)$, or a non-linear arbitrary function in the general case. As a shorthand, we use $\mathbf{x}_t = \mathbf{x}(s_t)$.

Usually, the true model $\mathcal{P}^\star$ and reward function $r^\star$ are not known to the agent. Instead, the agent interacts with the environment to collect samples and updates value prediction estimates:

$$\mathbf{w}_{t+1} = \mathbf{w}_t + \underbrace{\alpha \left[Y_t - v_{\mathbf{w}_t}(S_t)\right] \nabla_{\mathbf{w}_t} v_{\mathbf{w}_t}(S_t)}_{\equiv \Delta \mathbf{w}_t}, \qquad \text{(TD update)} \qquad (1)$$

where $Y_t$ is an *update target*. For instance, we could use Monte Carlo returns $Y_t = G_t$, or *temporal difference (TD) errors* (Sutton, 1988) $Y_t - v_{\mathbf{w}_t}(S_t) = \delta_t \equiv R_{t+1} + \gamma v_{\mathbf{w}_t}(S_{t+1}) - v_{\mathbf{w}_t}(S_t)$.

For control, an optimal action-value function $q_{\mathbf{w}}$ can be learned using Q-learning (Watkins and Dayan, 1992) updates: $\mathbf{w}_{t+1} = \mathbf{w}_t + \alpha \left[ R_{t+1} + \gamma \max_a q_{\mathbf{w}}(S_{t+1}, a) - q_{\mathbf{w}}(S_t, A_t) \right] \nabla_{\mathbf{w}_t} q_{\mathbf{w}_t}(S_t)$.

A MBRL agent learns an estimate $\mathcal{P}$ of the true model $\mathcal{P}^\star$ and $r$ of the reward function $r^\star$, a process called *model learning*. The agent can then employ a planning algorithm, that uses additional computation without additional experience to improve its predictions and/or behaviour. Usually, `Planner` uses models that look forward in time and anticipate a future state and reward conditioned on their input. A different option is a retrospect $\overleftarrow{\texttt{Planner}}$, which uses models that look backward in time and predict a predecessor state and corresponding reward.

Conventional approaches to model learning focus on learning good predictors of the environment, and ignore how `Planner` uses the model. Limited capacity and sampling noise can lead to model approximation errors, and the model can potentially choose to pay attention to aspects of the environment irrelevant to `Planner` (e.g., trying to predict a noisy TV). To mitigate this, value-aware model learning methods (Farahmand et al., 2017; Farahmand, 2018; Silver et al., 2017; Ayoub et al., 2020) attempt to find a model such that performing value-based planning on $v$ has a similar effect to applying the true environment model. Policy-aware model learning methods (Abachi et al., 2020) similarly look at the effect of planning on the policy, rather than values. In both cases, this means the model can focus on the aspect most important for the associated planning algorithm.

## 3 Planning Backward in Time

As a **thought experiment**, consider a simple model that looks forward or backward for one time-step to predict the next or the previous state. An agent takes action $a$ in $s$ and transitions to $s'$, experiencing a TD-error that changes the value prediction for $s$. To propagate this information backward to a predecessor state $\tilde{s}$ of $s$, forward models can face difficulties, because finding a good predecessor is nontrivial, and model misspecifications can cause a damaging update, pushing the value prediction estimate of a real state further away from its true value.

---

**Algorithm 1:** Backward Planning

1: **Input** policy $\pi, n$
2: $s \sim \text{env}()$
3: **for** each interaction $\{1, 2 \ldots T\}$ **do**
4:     $a \sim \pi(s)$
5:     $r, \gamma, s' \sim \text{env}(a)$
6:     $\overleftarrow{\mathcal{P}}, \overleftarrow{r} \leftarrow \text{model\_learning\_update}(s, a, s')$
7:     $v \leftarrow \text{learning\_update}(s, a, r, \gamma, s')$
8:     **for** each planning step $\{1, 2 \ldots N\}$ **do**
9:        $\tilde{s} \sim \overleftarrow{\mathcal{P}}(s), \tilde{r} \sim \overleftarrow{r}(\tilde{s}, s)$
10:      $v \leftarrow \overleftarrow{\texttt{Planner}}(\tilde{s}, \tilde{r}, \gamma, s)$
11:     $s \leftarrow s'$

---

Dyna-style planning methods (Sutton, 1990a) perform credit assignment by planning forward from previously visited states (or hypothetical states). This requires additional search-control and prioritization mechanisms. Otherwise: (i) the sampled state might be unrelated to the current state whose estimate has recently been updated; (ii) if the model is poor, planning steps can update the value of a real state with an erroneous imagined transition.

Backward models naturally sidestep these issues: (i) they can directly predict predecessor states $\tilde{s}$ of a newly updated state $s$; (ii) if the planning update of the imagined state $\tilde{s}$ solely uses $s$ as target for the update, a poor model will only damage the value prediction estimate of a *fictitious* state $\tilde{s}$.

We start our analysis with a treatment of backward planning which, in contrast to forward planning, operates using models running *backward in time*. We may write $\mathcal{P}_\pi^\star(s_{t+1}, a_t | s_t)$ in place of $\mathcal{P}_\pi^\star(S_{t+1} = s_{t+1}, A_t = a_t | S_t = s_t)$ in the interest of space.

**Assumptions** Throughout the paper, we make a stationarity assumption: for any policy $\pi$, the Markov chain induced by $\pi$, $\mathcal{P}_\pi^\star(s_{t+1}|s_t) = \sum_{a \in \mathcal{A}} \mathcal{P}^\star(s_{t+1}|s_t, a) \pi(a|s_t)$, is irreducible and aperiodic. We denote by $d_{\pi,t}(s)$ the probability of observing state $s$ at time $t$ when following $\pi$. Under the ergodicity assumption, each policy $\pi$ induces a unique stationary distribution of observed states $d_\pi(s) = \lim_{t \to \infty} d_{\pi,t}(s)$, as well as a stationary joint state-action distribution $d_\pi(s, a) = \pi(a|s) d_\pi(s)$.

**Backward models** A *backward transition model* identifies predecessor states of its input state. In formalizing *backward models*, we highlight some interesting properties (for which we defer details to

appendix A). To begin, *backward models are tethered to a policy*. Formally, we use $\overleftarrow{\mathcal{P}}^{\star}_{\pi,t}$ to refer to the dynamics of the *time-reversed* Markov chain induced by a policy $\pi$ at time-step $t$:

$$\overleftarrow{\mathcal{P}}^{\star}_{\pi,t}(s_t, a_t | s_{t+1}) = d_{\pi,t+1}(s_{t+1})^{-1} d_{\pi,t}(s_t) \pi(a_t | s_t) \mathcal{P}^{\star}(s_{t+1} | s_t, a_t) \tag{2}$$

and define $\overleftarrow{\mathcal{P}}^{\star}_{\pi}(s_t, a_t | s_{t+1}) \equiv \lim_{t \to \infty} \overleftarrow{\mathcal{P}}^{\star}_{\pi,t}(s_t, a_t | s_{t+1})$. One might hope that action-conditioning would relieve this policy dependence. Alas, it does not. An action-conditioned backward model for policy $\pi$ is defined as:

$$\overleftarrow{\mathcal{P}}^{\star}_{\pi}(s_t | s_{t+1}, a_t) = \frac{\pi(a_t | s_t)}{\bar{\pi}(a_t | s_{t+1})} \frac{d_{\pi}(s_t)}{d_{\pi}(s_{t+1})} \mathcal{P}^{\star}(s_{t+1} | s_t, a_t), \tag{3}$$

where $\bar{\pi}(a_t | s_{t+1})$ is the marginal probability of an action knowing the future state.

**Time-extended backward models**  Policy-conditioned models hold many shapes and have the potential to be useful in reasoning over larger timescales. Specifically, given a backward transition model $\overleftarrow{\mathcal{P}}^{\star}_{\pi}$, we define $\overleftarrow{\mathcal{P}}^{\star(n)}_{\pi}(\cdot|s) = (\overleftarrow{\mathcal{P}}^{\star}_{\pi})^n(\cdot|s)$ as the predecessor state distribution of having followed policy $\pi$ for $n$ steps, arriving at state $s$. Similarly, we denote the associated $n$-step reward model as: $\overleftarrow{r}^{\star(n)}_{\pi}(\tilde{s}, s) = \mathbb{E}\left[\sum_{t=0}^{n-1} \gamma^t R_{t+1} | S_0 = \tilde{s}, S_n = s, A_{t+1} \sim \pi(\cdot|S_t)\right]$. Other time-extended variants exist, such as $\lambda$-models (Sutton, 1995) or option models (Precup et al., 1998; Sutton et al., 1999), and could be learned counter-factually (Sutton et al., 2011) using an *excursion formulation* (Mahmood et al., 2015; Sutton et al., 2016; Zhang et al., 2020a,b; Gelada and Bellemare, 2019; Hallak and Mannor, 2017). We defer investigation of the off-policy regime to future work.

We primarily center on the prediction setting, in which the goal is to evaluate a given $\pi$, and simplify notation by removing the policy subscript from models and value functions.

*Backward models* are a pair of reward and transition models: $(\overleftarrow{\mathcal{P}}, \overleftarrow{r})$ (single or multi-step, and policy-dependent). The reward model takes in two endpoint states and outputs the estimated reward. Depending on its class, the transition model can output *a distribution of predecessor states*, *a sample predecessor* or *an expectation over prior states of its input*.

**Backward planning**  The hindsight planning mechanism we consider uses a *backward model* to identify the predecessor states $\tilde{s}$ of a particular state $s$. $\overleftarrow{\texttt{Planner}}$ projects backward in time, and from the projected states, performs forward-looking TD updates that end back in $s$. These corrections are used to re-evaluate the value predictions at states $\tilde{s}$. Such updates attempt to do credit assignment counter-factually by making parameter corrections in hindsight, given the new information gathered at the current step (the TD error of the transition $s \xrightarrow{a} s'$). Forward view corrections can be reformulated as backward corrections under the backward Markov chain (appendix B). For instance, an $n$-step learning update from any state $s$ can be formulated as:

$$\overleftarrow{\Delta}(s) = \mathbb{E}\left[\left(Y^{(n)}(S_{t-n}, S_t) - v_{\mathbf{w}}(S_{t-n})\right) \nabla_{\mathbf{w}} v_{\mathbf{w}}(S_{t-n}) | S_t = s, S_{t-n} \sim \overleftarrow{\mathcal{P}}^{(n)}(\cdot|S_t = s)\right], \tag{4}$$

where $Y^{(n)}(S_{t-n}, S_t) = \overleftarrow{r}^{\star}(S_{t-n}, S_t) + \gamma^n v_{\mathbf{w}}(S_t)$ is the n-step update target. For simplicity, in the following we use single-step models, and henceforth drop the horizon reference from the notations. Algorithm 1 sketches the generic steps of hindsight planning with backward models in a full framework of simultaneous learning and planning.

**Model estimation**  The choice for model estimation instantiates the above-mentioned algorithmic template. The most explicit way of computing $\overleftarrow{\Delta}$ is by learning *full probability distributions* – i.e. estimating the backward distribution $\overleftarrow{\mathcal{P}}(\cdot|s)$. Then, one can either (i) sample the model $\tilde{s} \sim \overleftarrow{\mathcal{P}}(\cdot|s)$ and do a stochastic update (or many): $\mathbf{w} = \mathbf{w} + \alpha \left(\overleftarrow{r}(\tilde{s}, s) + \gamma v_{\mathbf{w}}(s) - v_{\mathbf{w}}(\tilde{s})\right) \nabla_{\mathbf{w}} v_{\mathbf{w}}(\tilde{s})$, or (ii) perform a *distributional* backward planning update $\forall \tilde{s} \in \mathcal{S}$ in proportion to the probability given by the backward distribution model: $\mathbf{w} = \mathbf{w} + \alpha \overleftarrow{\mathcal{P}}(\tilde{s}|s)\left(\overleftarrow{r}(\tilde{s}, s) + \gamma v_{\mathbf{w}}(s) - v_{\mathbf{w}}(\tilde{s})\right) \nabla_{\mathbf{w}} v_{\mathbf{w}}(\tilde{s})$. In the general case, learning a full distribution model over the feature space is intractable. Alternatively, one can learn a backward *generative* model, sample predecessor features $\tilde{\mathbf{x}} \sim \overleftarrow{\mathcal{P}}(\cdot|\mathbf{x})$ and do one or more *sample* backward planning updates. We would like to think that maybe in the *linear* setting, where the gradient has the special form $\nabla_{\mathbf{w}} v_{\mathbf{w}}(\tilde{\mathbf{x}}) = \tilde{\mathbf{x}}$, one can get away with learning backward *expectation* models over features, and then perform an *expected* backward planning update. We

find however that a direct counterpart of the forward expectation models is not a valid update, as it involves a product of two (possibly) dependent random variables (the TD error and the gradient of the value function evaluated at the predecessor features given by the model). However, learning an unusual type of model still results in valid parameter corrections:

$$\mathbf{w} = \mathbf{w} + \alpha \left( \overleftarrow{r}_{\mathbf{x}}(\mathbf{x}) + \left( \gamma \overleftarrow{\mathcal{P}}_{\mathbf{x}}(\mathbf{x})\mathbf{x}^{\top} - \overleftarrow{\mathcal{P}}_{\mathbf{x}^2}(\mathbf{x}) \right) \mathbf{w} \right), \tag{5}$$

where $\overleftarrow{\mathcal{P}}_{\mathbf{x}}(\mathbf{x}) = \mathbb{E}[\tilde{\mathbf{x}}|\mathbf{x}]$ is a backward expectation model, $\overleftarrow{\mathcal{P}}_{\mathbf{x}^2}(\mathbf{x}) = \mathbb{E}[\tilde{\mathbf{x}}\tilde{\mathbf{x}}^{\top}|\mathbf{x}]$ is a covariance matrix of the predecessor features and $\overleftarrow{r}_{\mathbf{x}}(\mathbf{x}) = \mathbb{E}[\tilde{\mathbf{x}}\tilde{\mathbf{x}}^{\top}|\mathbf{x}]\Theta_r\mathbf{x}$ is a vector reward model with parameters $\Theta_r$ (appendix C). Note that this model requires estimating three quantities.

There are several approaches to estimating $\overleftarrow{\mathcal{P}}$, which can be characterized based on the constraints that they impose on the model. The standard approach is Maximum Likelihood Estimation (MLE) (appendix C): $\overleftarrow{\mathcal{P}} \leftarrow \operatorname{argmin}_{\overleftarrow{\mathcal{P}}^{\dagger} \in \mathcal{P}} \frac{1}{n} \sum_{S_i \in \mathcal{D}_n} \log \overleftarrow{\mathcal{P}}^{\dagger}(S_i)$, where we used $\mathcal{P}$ to denote the model space and $\mathcal{D}_n = \{(S_i, A_i, R_i, S_i')\}_{i=1}^n$ to represent collected data from interaction. Learning $\overleftarrow{\mathcal{P}}$ that minimizes a negative-log loss or other probabilistic losses leads to a model that tries to estimate all aspects of the environment. Estimating the reward model $\overleftarrow{r}$ defaults to a regression problem.

If however the true model does not belong to the model estimator's space and approximation errors exist, a planner-aware method can choose a model with minimum error with respect to $\overleftarrow{\texttt{Planner}}$'s objective. Both forward and backward planning objectives for value-based methods try to find an approximation $v$ to $v_\pi$ by applying one step of semi-gradient model-based TD update. A planner-aware model-learning objective is less constrained than the MLE objective in that it only tries to ensure that replacing the true dynamics with the model is inconsequential for the internal mechanism of $\overleftarrow{\texttt{Planner}}$. In the extreme case, we note that one can potentially directly parametrize and estimate the expected parameter corrections or updates, thus learning a fully *abstract model*. Learning of this kind shadows the internal arrow of time of the model. The ultimate unconstrained objective could meta-learn the model, such that, after a model-learning update, the model would be useful for planning. We offer some directions for planner-aware model learning in appendix C and defer an in-depth investigation of such methods to future work.

## 4 Empirical Studies

Our empirical studies work to uncover the distinctions between planning in anticipation and in retrospect. With the aim of understanding the underlying properties of these approaches, we ask the following questions:

*(i) How are the two planning algorithms distinct? When does it matter?*

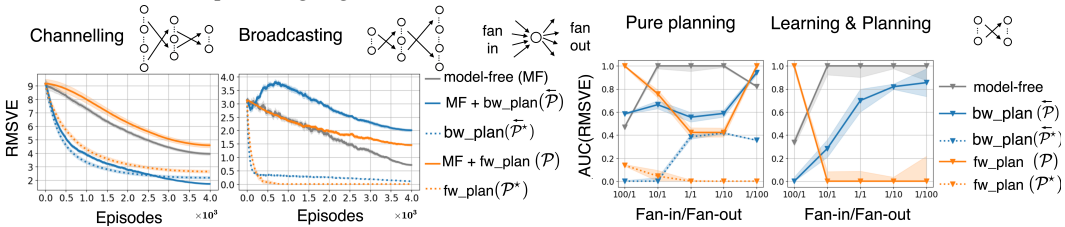

Figure 1: **(Left, Center-Left) Complementary properties of forward and backward planning:** Backward models work well in *channeling* structures, with large fan-in and small fan-out, while forward models are better suited for *broadcasting* state formations. The y-axis shows the RMSVE: $\sqrt{(|v_\pi - v|_2^2)}$; **(Right, Center-Right) Inflection point:** As we shift from channeling patterns (left) to broadcasting ones (right), the gain from one type of planning switches to the other, for both true and learned models. The y-axis shows the area under the curve (AUC) of the RMSVE (results are normalized by zero-centering and re-scaling by the $\max - \min$).

To understand which structural attributes of the environment are important for the two mechanisms and how such aspects interact with planning in online prediction settings we use the following experimental setup.

**Experimental setup** We explore the first question in a prediction setting using Markov Reward Processes where the states are organised as bi-graphs with with one (or more) sets of states (or levels)

$\{x_i\}_{i \in [0:n_x]}$ and $\{y_j\}_{j \in [0:n_y]}$ (Fig. 2), where we vary $n_x$ and $n_y$ in our experiments. We additionally experiment with adding intermediary levels: $\{z_k^l\}_{k \in [0:n_z^l], l \in [1:L]}$, where $L$ is the number of levels and $n_z^l$ is the size of level $l$. The states from a particular level transition only to states in the next level, thus establishing a particular flow and stationary structure of the Markov Chain under study.

We refer to the number of predecessors/successors a state might have in the state space as fan-in/fan-out. The experiments are ablation studies of the effects of varying the *fan-in* ($n_x$), the *fan-out* ($n_y$) and the number of levels $l$ with their corresponding sizes $n_z^l$.

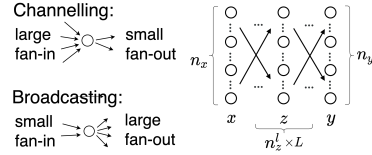

Figure 2: **Random Chain**: Illustration of the Markov Reward Process used in the prediction experiments. The chain flows from left to right.

For this investigation we performed two types of experiments: (i) on 3-level bipartite graphs as illustrated in figure 1-Left,Center-Left (ii) on 2-level bipartite graphs as shown in figure 1-Center-Right, Right. For (i), thumbnails depicts the phenomena of transitioning from a larger number of predecessors that funnel into a smaller number of successors, and vice-versa. The channelling pattern has the attributes: $L = 1, n_x = 500, n_z^l = 50, n_y = 5$, which are opposite from the broadcasting version: $L = 1, n_x = 5, n_z^1 = 50, n_y = 500$. For (ii) the results reported are for $(n_x, n_y) \in \{(500, 5), (50, 5), (5, 5), (5, 50), (5, 500)\}$, where we labeled the x-axis with the simplified ratio.

**Algorithms**    For the purposes of this experiment we use backward planning for value prediction. We include a complete pseudo-code for the backward planning algorithm used for this experiment in Algorithm 4, appendix D. For any transition $s \xrightarrow{a} s'$, we use the following reference frame to plan from: for backward models – we use the current state $s'$ of a transition, whereas for forward models – we use the previous state $s$ of a transition. The exact definition of reference frames is given in the following question we explore, corresponding to the next experiment.

**Context & observations**    Our studies identify an interesting phenomenon: the gain of the two planning mechanisms is reliant on two state attributes, which we call *fan-in* and *fan-out*. We illustrate this observation in the prediction setting presented above, depicted in the diagrams of Fig 1 and detailed in appendix D. We observe that large fan-in and small fan-out is better suited for backward planning, since backward planning updates many prior states at once (due to the large fan-in) and in these settings, due to small fan-out, backward models propagate lower-variance updates – see Fig. 1-Left. Intuitively, when many trajectories end up with the same outcome, all prior states' values can be updated with the new information available at the current state. This pattern, which we call *channeling*, also abates in part the vulnerability of backward models in updating states in a *sample-based* manner (i.e. states from which we correct predictions use a single sample, instead of the whole expectation as is the case for forward models). In contrast, forward models fit better a *broadcasting* formation (Fig. 1-Center-Left)). A backward model for this regime would be closer to factual TD and less efficient. Its predicted past states would need updates from many different successor states to construct accurate predictions. As we shift from the pattern of large fan-in/small fan-out to the opposite end, we notice a shift in the performance of the two planning mechanisms (Fig. 1-Right, Center-Right).

**Implications**    These results highlight one aspect of the problem central to the success of planning: the breadth of backward and forward projections; namely, we find anticipation to be sensible when the future is wide-ranging and predictable, and hindsight to work well when new discoveries affect many prior beliefs with certainty and to a great extent. Concurrently, these insights lay the groundwork for the development of new planning algorithms that dynamically choose where to plan *to* and *from*, seamlessly blending forethought and hindsight.

*(ii) Does it matter where the agent plans from? What is the effect of shifting the frame of reference used in planning?*

**Experimental setup**    In this experiment, as well as the following one, we perform ablation studies on the discrete navigation task from (Sutton and Barto, 2018) illustrated in Fig. 4 (details in (appendix D).

**Algorithms**    We operate in the control setting, for which we describe the algorithms we use, *Online Forward-Dyna* and *Online Backward-Dyna* (similar in nature to Sutton, 1990b; van Hasselt et al.,

2019) in algorithms 2 and 3, respectively (details in appendix D). In brief, both algorithms interlace additional steps of model learning and planning in-between steps of model-free Q-learning [1].

**Context & observations** We now ask whether the frame of reference (input state of `Planner` and $\overleftarrow{\texttt{Planner}}$, respectively), from which the agent starts planning, matters and if so, why. More precisely, consider a transition $s \xrightarrow{a} s'$ and note that we could use either $s$ or $s'$ as input to each planning algorithm. To show the effects of changing this frame of reference, we consider the control setting described at the beginning of the section and compare the action-value function variants that employ each of the planning mechanisms proposed, namely *Online Forward-Dyna* and *Online Backward-Dyna* ( appendix D for details).

| **Algorithm 2:** Online Forward-Dyna: Learning, Acting & Forward Planning | **Algorithm 3:** Online Backward-Dyna: Learning, Acting & Backward Planning |
|---|---|
| 1: **Input** policy $\pi, n$ | 1: **Input** policy $\pi, n$ |
| 2: $s \sim \text{env}()$ | 2: $s \sim \text{env}()$ |
| 3: **for** each interaction $\{1, 2 \dots T\}$ **do** | 3: **for** each interaction $\{1, 2 \dots T\}$ **do** |
| 4:    $a \sim \text{argmax}_a\, q(s, a)$ | 4:    $a \sim \text{argmax}_a\, q(s, a)$ |
| 5:    $r, \gamma, s' \sim \text{env}(a)$ | 5:    $r, \gamma, s' \sim \text{env}(a)$ |
| 6:    $\mathcal{P}, \overrightarrow{r}, \overrightarrow{\gamma} \leftarrow \text{model\_learning\_update}(s, a, s')$ | 6:    $\overleftarrow{\mathcal{P}}, \overleftarrow{r} \leftarrow \text{model\_learning\_update}(s, a, s')$ |
| 7:    $q \leftarrow \text{learning\_update}(s, a, r, \gamma, s')$ | 7:    $q \leftarrow \text{learning\_update}(s, a, r, \gamma, s')$ |
| 8:    $s_{\text{ref}} \leftarrow \text{planning\_reference\_state}(s, s')$ | 8:    $s_{\text{ref}} \leftarrow \text{planning\_reference\_state}(s, s')$ |
| 9:    **for** each $a \in \mathcal{A}$ **do** | 9:    **for** each $\tilde{s} \in \mathcal{S}, \tilde{a} \in \mathcal{A}$ **do** |
| 10:      **for** each $s' \in \mathcal{S}$ **do** | 10:      $y = \overleftarrow{r}(s_{\text{ref}}) + \gamma \max_{\bar{a}} q(s_{\text{ref}}, \bar{a})$ |
| 11:        $y = \overrightarrow{r}(s') + \overrightarrow{\gamma}(s') \max_{a'} q(s', a')$ | 11:      $\overleftarrow{\Delta}(\tilde{s}, \tilde{a}) = \overleftarrow{\mathcal{P}}(\tilde{s}, \tilde{a}\|s_{\text{ref}}) \left(y - q(\tilde{s}, \tilde{a})\right)$ |
| 12:        $\Delta(s_{\text{ref}}, a) \leftarrow \Delta(s_{\text{ref}}, a) +$ $\mathcal{P}(s'\|s_{\text{ref}}, a)\left(y - q(s_{\text{ref}}, a)\right)$ | 12:      $q(\tilde{s}, \tilde{a}) \leftarrow q(\tilde{s}, \tilde{a}) + \alpha\overleftarrow{\Delta}(\tilde{s}, \tilde{a})$ |
| 13:      $q(s_{\text{ref}}, a) \leftarrow q(s_{\text{ref}}, a) + \alpha\Delta(s_{\text{ref}}, a)$ | 13:    $s \leftarrow s'$ |
| 14:    $s \leftarrow s'$ | 14: |

We compare the pure model-based setting and the full learning framework (learning & planning). In the full learning framework shown in Fig. 3-Left, planning backward from $s$ implies the use of new knowledge about the current prediction, as we bootstrap on the value at $s$ that has recently been updated: $\max_{\bar{a}} q(s, \bar{a})$; in contrast, applying planning from $s'$ achieves a somewhat different effect: it complements model-free learning (as it bootstraps on $q(s', \cdot)$, which has not changed, but may still benefit from the reward information $r(s')$).

Contrary to backward planning, the forward counterpart gains from being as anticipatory as possible, by planning from the current state $s'$. The effects are reversed in the pure planning setting (Fig 3-Right). Specifically, the backward model cannot rely on model-free learning to re-evaluate predecessor predictions with values at $s$, since these are no longer changed by the learning process; it thus assumes that role. Simultaneously, backward planning is more efficient at state $s'$ since is benefits from the current additional transition $s \xrightarrow{a} s'$. Likewise, forward planning is more reliable in $s$ by the same argument, and also assumes the role of *learning*.

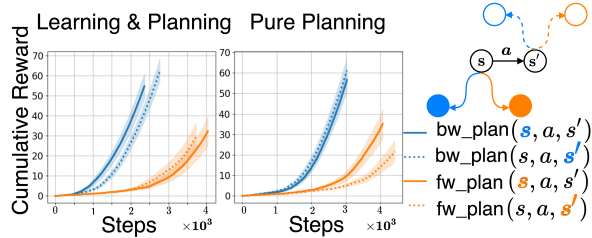

Figure 3: **Planning frame of reference**: **(Left)** In the full learning setting (learning and planning), the agent is more effective by planning backward from $s$ and planning forward from $s'$. **(Right)** In the pure planning setting, both planning mechanisms assume the role of learning and gain more by processing the exact same opposite states of the full learning case (Left), remaining in antithesis.

**Implications** These results emphasize that both planning mechanisms work best when they complement model-free learning, if it is used, and both can take on its role, if it is not.

*(iii) How is planning influential on behaviour?*

**Experimental setup**   This experiment is done in the same control setting described above[2].

**Context & observations**   Our results provide evidence for the following observations: (i) errors in the backward model, caused by stochastic transitions, are to a lesser extent damaging for credit assignment (Fig. 4 Top-Right); (ii) backward planning accelerates the search for the optimal policy in the presence of stochastic rewards (Fig. 4-Bottom-Left and Bottom-Right); (iii) for extremely stochastic rewards, even backward models fail to capture the dynamics accurately enough (Fig. 4 Bottom-Right); (iv) model misspecification affects to a deeper extent forward planning.

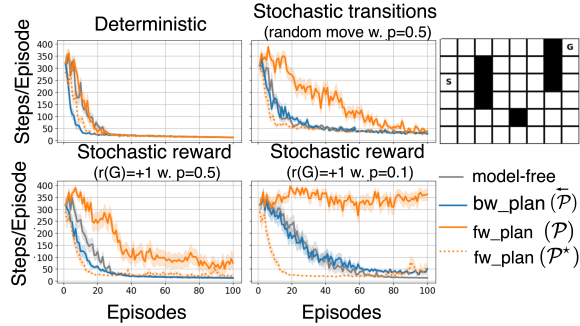

Figure 4: **Information propagation in stochastic settings**: backward models can propagate new information faster in stochastic reward settings and are more robust to randomness in the dynamics. Planning with the true forward model emphasizes the issues with forward planning (planning with the true backward model is omitted, as it depends on the constantly changing policy).

**Implications**   These results emphasize the potential impact, in environments with high stochasticity, of a different pattern of reasoning, related to counterfactual learning. Particularly, an agent can *project back* to a potential causal state and *rethink* its decisions after each new experience. More investigation of this idea would be useful.

## 5   Related Work

Backward models and prioritized sweeping (Moore and Atkeson, 1993; Peng and Williams, 1993; McMahan and Gordon, 2005; Sutton et al., 2012; van Seijen and Sutton, 2013) have been used within the Dyna framework (Sutton, 1990a,b), mostly in combination with heuristic strategies for ordering backups. Other search control strategies have explored the use of non-parametric models for prioritizing experience (Schaul et al., 2016; Pan et al., 2018). The experiments of van Hasselt et al. (2019) with backward models for credit assignment have motivated our study; we worked to understand these issues in depth, extending and complementing their results. Concurrently with our work, Jafferjee et al. (2020) also builds on the work of van Hasselt et al. (2019), and investigates model misspecification for backward planning with linear models.

Goyal et al. (2019) proposes predicting predecessor states on a trajectory, similarly to the ideas of episodic control (Lengyel and Dayan, 2007). Their approach uses imitation learning on a generative model's outputs to improve exploration by incentivizing the agent towards the high value states on which the model was trained. In contrast, we aim to formalize and tease apart the fundamental properties of online hindsight planning. Schroecker et al. (2019) also applies generative predecessor models for imitation learning with policy gradient methods.

Interestingly, research reveals that planning in the brain might also involve two processes that resemble forward and backward planning (Mattar and Daw, 2018). Other research into credit assignment with hindsight reasoning, uses either a structural causal model to reason about the causes of events from trajectories (Buesing et al., 2019) or an inverse dynamics model conditioned on future outcomes to determine the relevance of past actions to particular outcomes (Harutyunyan et al., 2019b). The temporal value transport of Hung et al. (2019) uses an attention mechanism over an external memory to jump over irrelevant parts of trajectories and propagate credit backward in time. Ke et al. (2018) similarly attempts to propagate credit backward, yet their approach is to use an RNN with learned associative connections to do so.

Pitis (2019) addresses the problem of credit assignment by proposing a tabular version of cumulative predecessor representations. van Hasselt et al. (2020) expands this to the general case of expected

eligibility traces. Satija et al. (2020) use backward value functions to encode constraints for solving constrained MDPs (CMDPs) with safe policy improvements. Though both our work and theirs employ some form of retrospective knowledge, the contents, purposes and uses differ. The concurrent work of Zhang et al. (2020b) also utilizes backward value functions, albeit for anomaly detection. Harutyunyan et al. (2019a) also connects model entropy and fan-in, although their aim is option discovery.

Model learning objectives and strategies that ground model prediction in estimates other than environment observations have been been considered in Joseph et al. (2013); Farahmand et al. (2017); Silver et al. (2017); Farahmand (2018); Oh et al. (2017); Farquhar et al. (2017); D'Oro et al. (2019); Luo et al. (2019); Schrittwieser et al. (2019); Abachi et al. (2020); Ayoub et al. (2020).

## 6 Closing and Future Work

We explored several questions pertaining to the nature, use and attributes of planning. We provided motivation, formalism and insight for hindsight planning, emphasizing the properties of backward models. Particularly, we looked at how forward and backward planning exhibit complementary gains in opposite settings. Further, we highlighted a spectrum of model learning objectives that increasingly add more flexibility. Lastly, we performed ablation studies that reveal interesting properties about the nature of planning, their instruments – models, and the context in which they operate.

The key takeaways from our work are: (i) the problem dimension of the transition dynamics, resulting from the world dynamics and the agent's policy is of great importance for the effectiveness of planning; we demonstrated the two planning mechanisms exhibit complementary properties – if the future is broad and predictable, forethought is invaluable, if backward hypotheses are causal and less determined by chance, hindsight planning is effective; (ii) planning behaves differently in anticipation and in retrospect; (iii) the states we pick to plan from matter, and the best states to consider for forward and backward planning differ; (iv) planning is complementary to model-free updates, and should be aware of these updates — for instance, the best use of planning can depend on which states are being updated model-free; (v) backward planning can be favorable over forward planning in stochastic environments by proposing a different pattern of reasoning, related to counterfactual learning resulting in more efficient credit assignment.

Planning is a very broad topic, and much interesting work remains to be done; we name some directions that seem most exciting to us. The interaction of planning with channelling and broadcasting patterns induces interesting questions in relation to time-extended models, for instance in terms of choosing where to plan from and to (e.g., cf. Harutyunyan et al., 2019a). Planning backwards with time-extended models has received little attention in the literature, but appears promising. Equally, these insights pave the way for future work in developing new planning algorithms that take advantage of this property to seamlessly and optimally integrate both forethought and hindsight. Lastly, we conjecture a potential virtuous cycle of backward planning and generalization, in which the former can complement the latter, either by connecting areas of the latent space where generalization is poor or, ideally, by contributing to the construction of a better representation. Generalization, in turn, can broaden the effect of backward planning through abstraction.

### Broader Impact

Our work deals with fundamental insights related to the nature of model-based reinforcement learning. The problem of building models of the world and planning with them to achieve desired objectives is of paramount importance for real world applications of intelligent systems. However, in this work, we do not focus on applications, but instead look at the problem from a theoretical and investigative angle and largely treat it conceptually. As such, we consider this not to be applicable in this setting.

## Footnotes

[1]N.B. despite the tabular setting, learning is online and planning uses parametric models only in reference to the current transition. This is because we are interested in insights that transfer to more complex environments.

[2]For readers familiar with standard Dyna results, we note that our setting differs from the conventional Dyna setting, as do our results. Whereas classically, tabular Dyna-agents assume access to the entire state space when deciding where to plan from, we deliberately do not make this assumption.

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
