[Supplementary Material]

# A   Theoretical Background and Derivations

In the following, we use $\breve{\mathcal{P}}_\pi(s_{t-1}, a_{t-1}|s_t)$ as a shorthand for $\breve{\mathcal{P}}_\pi(S_{t-1} = s_{t-1}, A_{t-1} = a_{t-1}|S_t = s_t)$ to unclutter notation and drop the $\star$ superscript in places where it is obvious we refer to the true transition model.

## A.1   Backward Models are Policy Dependent

Using the stationarity assumption, we write $d_\pi(s)$ as the stationary distribution of $\pi$ at $s$. The joint posterior of the previous state and action can be expressed using Bayes rule as:

$$\breve{\mathcal{P}}_\pi(s_{t-1}, a_{t-1}|s_t) = \frac{\mathcal{P}(s_t|s_{t-1}, a_{t-1})\mathcal{P}_\pi(s_{t-1}, a_{t-1})}{\sum_{s\in\mathcal{S}}\sum_{a\in\mathcal{A}}\mathcal{P}(s_t|s, a)\mathcal{P}_\pi(s, a)}. \tag{6}$$

We can express the prior as: $\mathcal{P}_\pi(s_{t-1}, a_{t-1}) = \pi(a_{t-1}|s_{t-1})d_\pi(s_{t-1})$. The denominator can be folded into the stationary distribution of the policy at the next state: $\sum_{s\in\mathcal{S}}\sum_{a\in\mathcal{A}}\mathcal{P}(s_t|s, a)\mathcal{P}_\pi(s, a) = d_\pi(s_t)$. Then the joint model is

$$\breve{\mathcal{P}}_\pi(s_{t-1}, a_{t-1}|s_t) = \frac{\mathcal{P}(s_t|s_{t-1}, a_{t-1})\pi(a_{t-1}|s_{t-1})d_\pi(s_{t-1})}{d_\pi(s_t)} \tag{7}$$

## A.2   Action Conditioned Backward Models

Backward models mirror their counterparts – forward models. As such, one may define action-condition backward models, yet note that these too are dependent on a policy.

According to Bayes, the probability of a previous state $\breve{\mathcal{P}}(s_{t-1}|s_t, a_{t-1})$, conditioned on a *previous* action $a_{t-1}$ and policy $\pi$, leading into the current state $s_t$ is:

$$\breve{\mathcal{P}}_\pi(s_{t-1}|s_t, a_{t-1}) = \frac{\mathcal{P}(s_t|s_{t-1}, a_{t-1})\mathcal{P}_\pi(s_{t-1}|a_{t-1})}{\mathcal{P}_\pi(s_t|a_{t-1})} \tag{8}$$

Applying Bayes rule again for the probability of a previous state $\mathcal{P}_\pi(s_{t-1}|a_{t-1})$ conditioned on a previous action $a_{t-1}$ and policy $\pi$

$$\mathcal{P}_\pi(s_{t-1}|a_{t-1}) = \frac{\pi(a_{t-1}|s_{t-1})d_\pi(s_{t-1})}{\pi(a_{t-1})} \propto d_\pi(s_{t-1}, a_{t-1}) \tag{9}$$

The denominator can be rewritten using Bayes as

$$\mathcal{P}_\pi(s_t|a_{t-1}) = \frac{\overleftarrow{\pi}(a_{t-1}|s_t)d_\pi(s_t)}{\pi(a_{t-1})} \propto d_\pi(s_t, a_{t-1}) \tag{10}$$

The posterior probability, i.e. the action-conditioned backward model for policy $\pi$ becomes

$$\breve{\mathcal{P}}_\pi(s_{t-1}|s_t, a_{t-1}) = \frac{\mathcal{P}(s_t|s_{t-1}, a_{t-1})d_\pi(s_{t-1}, a_{t-1})}{d_\pi(s_t, a_{t-1})} \tag{11}$$

$$= \frac{\pi(a_{t-1}|s_{t-1})}{\overleftarrow{\pi}(a_{t-1}|s_t)}d_\pi(s_t)^{-1}d_\pi(s_{t-1})\mathcal{P}(s_t|s_{t-1}, a_{t-1}), \tag{12}$$

where $\overleftarrow{\pi}(a_{t-1}|s_t)$ gives the marginal probability of an action, conditioned on the future state $s_t$.

## A.3   Multi-step models

Policy-conditioned models hold many shapes and have the potential to be useful in reasoning over larger timescales. To start, given a backward transition model $\breve{\mathcal{P}}_\pi^\star$, we define $\breve{\mathcal{P}}_\pi^{\star(n)}(\cdot|s) = (\breve{\mathcal{P}}_\pi^\star)^n(\cdot|s)$ as the predecessor state distribution of having followed policy $\pi$ for $n$ steps, arriving at state $s$. Similarly, we denote the associated $n$ steps reward model as: $\overleftarrow{r}_\pi^\star(\tilde{s}, s) = \mathbb{E}\left[\sum_{t=0}^{n-1}\gamma^t R_{t+1}|S_0 = \tilde{s}, S_n = s, A_{t+1}\sim\pi(\cdot|S_t)\right]$. For a *single-step* model we can marginalize the actions from the joint predecessor and action model as: $\breve{\mathcal{P}}_\pi^\star(\tilde{s}|s) = \sum_{\tilde{a}}\breve{\mathcal{P}}_\pi^\star(\tilde{s}, \tilde{a}|s)$. An *n-step model* is one that predicts the probability of a predecessor state $n$-steps in the past.

A multi-time model is a combination of $n$-step models:

$$\breve{\mathcal{P}}_\pi^{\star(1:\infty)}(\tilde{s}|s) = \sum_{n=1}^{\infty} w_n\breve{\mathcal{P}}_\pi^{\star(n)}(\tilde{s}|s), \tag{13}$$

$$\overleftarrow{r}_\pi^{\star(1:\infty)}(\tilde{s}, s) = \sum_{n=1}^{\infty} w_n\overleftarrow{r}_\pi^{\star(n)}(\tilde{s}, s), \tag{14}$$

where $\sum_{n=1}^{\infty} w_n = 1$. In the general case, the number of steps we "remember" the past can be a random variable. We can augment the state space with an event indicating the decision to consider a state as predecessor (and to re-evaluate predictions therefrom) or to look further back in time. If the probability associated with the event is given by a function $\lambda : \mathcal{S} \to [0, 1]$, we can write the model with the semantics of continuing the backward process as $\breve{\mathcal{P}}_{\pi,\lambda}^{\star}(\tilde{s}|s) = \breve{\mathcal{P}}_{\pi}^{\star}(\tilde{s}|s)\lambda(\tilde{s})$, and denote backward bootstrapping with $1 - \lambda(\tilde{s})$ at state $\tilde{s}$ and re-evaluating predictions.

N-step models involve a hard cutoff contingent on the number of steps. If $\lambda$ is a smooth function the resulting models allow for a spectrum of time-extended models ranging from single-step models to n-step models and a wide variety in between. Having a $\lambda$ independent of state defaults to interpolating between $n$-step predecessors with a mixture:

$$\breve{\mathcal{P}}_{\pi,\lambda}^{\star}(\tilde{s}|s) = (1 - \lambda)\sum_{n=0}^{\infty} \lambda^n \breve{\mathcal{P}}_{\pi}^{\star(n)}(\tilde{s}|s), \tag{15}$$

Likewise, a state dependent $\lambda$-*model* is instrumental when the scale of prediction varies. Therefore, a backward model conditioned on the smooth state-dependent $\lambda$ function can choose to ignore the inconsequential chaos lying in-between cause and effect [3]. We can obtain recursive formulations for these models (similarly to Bellman equations):

$$\breve{\mathcal{P}}_{\pi,\lambda}^{\star}(\tilde{s}|s) = (1 - \lambda(\tilde{s}))\,\breve{\mathcal{P}}_{\pi}^{\star}(\tilde{s}|s) + \sum_{\tilde{\tilde{s}}} \lambda(\tilde{\tilde{s}})\breve{\mathcal{P}}_{\pi}^{\star}(\tilde{\tilde{s}}|s)\breve{\mathcal{P}}_{\pi,\lambda}^{\star}(\tilde{s}|s). \tag{16}$$

Subsuming all other formulations, *backward option models* may also be defined as a special type of backward value functions tethered, not just to a policy, but to a triple $(\pi_o, \iota_o, \beta_o)$, where $\pi_o$ is the decision policy, $\iota_o$ and $\beta_o$ determine the initiation and termination of the option. A *backward option model* is defined as:

$$\breve{\mathcal{P}}_{\pi_\mathcal{O}}^{\star}(\tilde{s}|s, o) = \iota_o(\tilde{s})\breve{\mathcal{P}}_{\pi_o}^{\star}(\tilde{s}|s) + \sum_{\tilde{\tilde{s}}} \left(1 - \iota_o(\tilde{\tilde{s}})\right)\breve{\mathcal{P}}_{\pi_o}^{\star}(\tilde{\tilde{s}}|s)\breve{\mathcal{P}}_{\pi_\mathcal{O}}^{\star}(\tilde{s}|\tilde{\tilde{s}}, o) \tag{17}$$

where $\iota(\tilde{s})$ denotes initiation at the predecessor state $\tilde{s}$ and $\pi_\mathcal{O}$ denotes a policy over options.

Extended time models could be learned on-policy by Monte-Carlo methods by taking samples of trajectories from the forward Markov chain induced by policy $\pi$, or by TD-learning using a recursive formulation. For instance, $\lambda$-models could be updated with the rule:

$$\breve{\mathcal{P}}_{\pi,\lambda}(\tilde{s}|s_t) = \breve{\mathcal{P}}_{\pi,\lambda}(\tilde{s}|s_t) + \alpha\left((1 - \lambda(s_t))\mathbf{1}_{s_{t-1}=\tilde{s}} + \lambda(s_t)\breve{\mathcal{P}}_{\pi,\lambda}(\tilde{s}|s_{t-1}) - \breve{\mathcal{P}}_{\pi,\lambda}(\tilde{s}|s_t)\right) \tag{18}$$

One can use any of these models in computing value functions, yet some types of models will have different properties. If we would have estimators for n-step models, $\forall n$, or a $\lambda$-model, notice that we can update backward all trajectories at once, in contrast to a single trajectory as is the case for eligibility traces (Sutton, 1988), and similarly to van Hasselt et al. (2020).

# B  Forward-Backward Equivalences

In this appendix we review some equivalences of forward and backward views and include for completeness some properties of forward and reverse Markov chains (cf. Morimura et al., 2010), along with other forward-backward equivalences (cf. Sutton, 1988).

## B.1  Forward-Backward Views – Equivalence of Temporal Difference Learning

The temporal difference learning over an episode of experience can be equivalently interpreted in the off-line regime looking forward or backward (cf. Sutton, 1988):

$$\mathbf{w} = \mathbf{w} + \alpha \sum_{t=0}^{\infty} \left[ \sum_{k=t}^{\infty} \gamma^{k-t} R_{k+1} - v_{\mathbf{w}}(S_t) \right] \nabla_{\mathbf{w}} v_{\mathbf{w}}(S_t) \qquad \text{(forward view)} \qquad (19)$$

$$= \mathbf{w} + \alpha \sum_{t=0}^{\infty} \sum_{k=t}^{\infty} \gamma^{k-t} \left[ R_{k+1} + \gamma v_{\mathbf{w}}(S_{k+1}) - v_{\mathbf{w}}(S_k) \right] \nabla_{\mathbf{w}} v_{\mathbf{w}}(S_t) \qquad (20)$$

$$= \mathbf{w} + \alpha \sum_{k=0}^{\infty} \sum_{t=0}^{k} \gamma^{k-t} \delta_k \nabla_{\mathbf{w}} v_{\mathbf{w}}(S_t) \qquad (21)$$

$$= \mathbf{w} + \alpha \sum_{t=0}^{\infty} \delta_t \sum_{k=0}^{t} \gamma^{t-k} \nabla_{\mathbf{w}} v_{\mathbf{w}}(S_k) \qquad \text{(backward view)}. \qquad (22)$$

## B.2  Relationship Forward-Backward Markov Chains

**Detailed balance**   The relationship between the forward and the backward model can be written as

$$\bar{\mathcal{P}}_{\pi}(s_{t-1}, a_{t-1}|s_t) = \frac{\mathcal{P}(s_t|s_{t-1}, a_{t-1})\pi(a_{t-1}|s_{t-1})d_{\pi}(s_{t-1})}{d_{\pi}(s_t)} \qquad (23)$$

$$= \frac{\mathcal{P}_{\pi}(s_t, a_{t-1}|s_{t-1})d_{\pi}(s_{t-1})}{d_{\pi}(s_t)}. \qquad (24)$$

Multiplying by $d_{\pi}(s_t)$ and summing over all actions $a_{t-1}$ in the previous equation gives the **detailed balance equation** (MacKay, 2002):

$$d_{\pi}(s_t)\bar{\mathcal{P}}_{\pi}(s_{t-1}|s_t) = \mathcal{P}_{\pi}(s_t|s_{t-1})d_{\pi}(s_{t-1}) \qquad (25)$$

**Stationary distribution equivalence**   Summing over all states $s_t$

$$\sum_{s_t \in \mathcal{S}} \bar{\mathcal{P}}_{\pi}(s_{t-1}|s_t)d_{\pi}(x_t) = d_{\pi}(s_{t-1}) \qquad (26)$$

we have $d_{\pi}$ also as the stationary distribution under the backward Markov chain, and consequently that the forward and backward chains have the same stationary distribution.

**Backward transition matrix**   The backward transition dynamics matrix can be written as

$$\bar{\mathbf{P}}_{\pi} = \text{diag}(d_{\pi})^{-1} \mathbf{P}_{\pi}^{\top} \text{diag}(d_{\pi}) \qquad (27)$$

## B.3  Equivalence between Forward and Backward Planning

The expected parameter corrections for n-step learning updates are defined over the forward Markov chain $\mathcal{F}$ induced by a policy $\pi$ and such can be redefined in terms of the backward Markov chain $\mathcal{B}$ as:

$$\mathbb{E}_{\mathcal{F}(\pi)} \left[ \left( \bar{r}^{\star(n)}(S_t, S_{t+n}) + \gamma v_{\mathbf{w}}(S_{t+n}) - v_{\mathbf{w}}(S_t) \right) \nabla_{\mathbf{w}} v_{\mathbf{w}}(S_t) | S_{t+n}, d_{\pi}(S_t) \right] = \qquad (28)$$

$$= \mathbb{E}_{\mathcal{F}(\pi)} \left[ \left( \bar{r}^{\star(n)}(S_{t-n}, S_t) + \gamma v_{\mathbf{w}}(S_t) - v_{\mathbf{w}}(S_{t-n}) \right) \nabla_{\mathbf{w}} v_{\mathbf{w}}(S_{t-n}) | S_t, d_{\pi}(S_{t-n}) \right] \qquad (29)$$

$$= \mathbb{E}_{\mathcal{B}(\pi)} \left[ \left( \bar{r}^{\star(n)}(S_{t-n}, S_t) + \gamma v_{\mathbf{w}}(S_t) - v_{\mathbf{w}}(S_{t-n}) \right) \nabla_{\mathbf{w}} v_{\mathbf{w}}(S_{t-n}) | S_t \right], \qquad (30)$$

since

$$\breve{\mathcal{P}}_\pi(s_{t-1}, a_{t-1}, \ldots, s_{t-n}, a_{t-n}|s_t) = \breve{\mathcal{P}}_\pi(s_{t-1}, a_{t-1}|s_t) \ldots \breve{\mathcal{P}}_\pi(s_{t-n}, a_{t-n}|s_{t-n+1}) \tag{31}$$

$$= \frac{\mathcal{P}_\pi(s_t, a_{t-1}|s_{t-1}) \ldots \mathcal{P}_\pi(s_{t-n+1}, a_{t-n}|s_{t-n}) d_\pi(s_{t-n})}{d_\pi(s_t)} \tag{32}$$

$$\propto \mathcal{P}_\pi(s_t, a_{t-1}|s_{t-1}) \ldots \mathcal{P}_\pi(s_{t-n+1}, a_{t-n}|s_{t-n}) d_\pi(s_{t-n}) \tag{33}$$

The backward parameter corrections can be defined over the backward distribution as:

$$\breve{\Delta}(s) = \mathbb{E}\left[\left(\breve{r}^{\star(n)}(S_{t-n}, S_t) + \gamma v_{\mathbf{w}}(S_t) - v_{\mathbf{w}}(S_{t-n})\right) \nabla_{\mathbf{w}} v_{\mathbf{w}}(S_{t-n})|S_t = s, S_{t-n} \sim \breve{\mathcal{P}}_\pi^{\star(n)}(\cdot|S_t = s)\right] \tag{34}$$

## C Model-Learning Objectives

### C.1 Expected Linear Backward Models

We assume the linear representation $v_\mathbf{w}(\mathbf{x}) = \mathbf{x}^\top \mathbf{w}$ for the value function. For the reward model we express the parameters of the reward using weights $\Theta_r \in \mathbb{R}^{d \times d}$, such that $\mathbf{x}(\tilde{s})^\top \Theta_r \mathbf{x}(s) \approx \mathbb{E}[R_t | \tilde{s} = S_{t-1}, s = S_t]$. The expected parameter corrections from some arbitrary features $\mathbf{x}$ can be written as:

$$\sum_{\tilde{\mathbf{x}}} Pr(\tilde{\mathbf{x}}|\mathbf{x}) \left( \sum_{\tilde{r}} Pr(\tilde{r}|\tilde{\mathbf{x}}, \mathbf{x})\tilde{r} + \gamma v_\mathbf{w}(\mathbf{x}) - v_\mathbf{w}(\tilde{\mathbf{x}}) \right) \tilde{\mathbf{x}} = \tag{35}$$

$$= \sum_{\tilde{\mathbf{x}}} Pr(\tilde{\mathbf{x}}|\mathbf{x}) \left( \tilde{\mathbf{x}}^\top \Theta_r \mathbf{x} + \gamma \mathbf{x}^\top \mathbf{w} - \tilde{\mathbf{x}}^\top \mathbf{w} \right) \tilde{\mathbf{x}} \tag{36}$$

$$= \sum_{\tilde{\mathbf{x}}} Pr(\tilde{\mathbf{x}}|\mathbf{x})\tilde{\mathbf{x}}\tilde{\mathbf{x}}^\top \Theta_r \mathbf{x} + \gamma \sum_{\tilde{\mathbf{x}}} Pr(\tilde{\mathbf{x}}|\mathbf{x})\tilde{\mathbf{x}}\mathbf{x}^\top \mathbf{w} - \sum_{\tilde{\mathbf{x}}} Pr(\tilde{\mathbf{x}}|\mathbf{x})\tilde{\mathbf{x}}\tilde{\mathbf{x}}^\top \mathbf{w} \tag{37}$$

$$= \mathbb{E}\left[\tilde{\mathbf{x}}\tilde{\mathbf{x}}^\top | \mathbf{x}\right] \Theta_r \mathbf{x} + \gamma \mathbb{E}\left[\tilde{\mathbf{x}}^\top | \mathbf{x}\right] \mathbf{x}^\top \mathbf{w} - \mathbb{E}\left[\tilde{\mathbf{x}}\tilde{\mathbf{x}}^\top | \mathbf{x}\right] \mathbf{w} \tag{38}$$

$$= \mathbb{E}\left[\tilde{\mathbf{x}}\tilde{\mathbf{x}}^\top | \mathbf{x}\right] \Theta_r \mathbf{x} + \left( \gamma \mathbb{E}\left[\tilde{\mathbf{x}}|\mathbf{x}\right] \mathbf{x}^\top - \mathbb{E}\left[\tilde{\mathbf{x}}\tilde{\mathbf{x}}^\top | \mathbf{x}\right] \right) \mathbf{w} \tag{39}$$

$$= \breve{r}_\mathbf{x}(\mathbf{x}) + \left( \gamma \breve{\mathcal{P}}_\mathbf{x}(\mathbf{x})\mathbf{x}^\top - \breve{\mathcal{P}}_{\mathbf{x}^2}(\mathbf{x}) \right) \mathbf{w}, \tag{40}$$

where $\breve{\mathcal{P}}_\mathbf{x}(\mathbf{x}) = \mathbb{E}\left[\tilde{\mathbf{x}}|\mathbf{x}\right]$ is a backward expectation model, $\breve{\mathcal{P}}_{\mathbf{x}^2}(\mathbf{x}) = \mathbb{E}\left[\tilde{\mathbf{x}}\tilde{\mathbf{x}}^\top | \mathbf{x}\right]$ is a predecessor features covariance matrix and $\breve{r}_\mathbf{x}(\mathbf{x}) = \mathbb{E}\left[\tilde{\mathbf{x}}\tilde{\mathbf{x}}^\top | \mathbf{x}\right] \Theta_r \mathbf{x}$ is a vector reward model ($\Theta_r$ are the parameters of the expected linear reward model).

In contrast, the forward view update is:

$$\sum_{\mathbf{x}'} \left( Pr(\mathbf{x}'|\mathbf{x}) \left( \sum_r Pr(r|\mathbf{x}, \mathbf{x}')r + \gamma v_\mathbf{w}(\mathbf{x}') \right) - v_\mathbf{w}(\mathbf{x}) \right) \mathbf{x} = \tag{41}$$

$$\left( \sum_{\mathbf{x}'} Pr(\mathbf{x}'|\mathbf{x})(\mathbf{x}')^\top \Theta_r \mathbf{x} + \gamma \sum_{\mathbf{x}'} Pr(\mathbf{x}'|\mathbf{x})(\mathbf{x}')^\top \mathbf{w} - \mathbf{x}^\top \mathbf{w} \right) \mathbf{x} \tag{42}$$

$$= \mathbb{E}\left[\mathbf{x}'|\mathbf{x}\right] \Theta_r \mathbf{x}\mathbf{x} + \left( \gamma \mathbf{x}\mathbb{E}\left[(\mathbf{x}')^\top|\mathbf{x}\right] - \mathbf{x}\mathbf{x}^\top \right) \mathbf{w} \tag{43}$$

$$= \mathbf{x} \left( \breve{r}_\mathbf{x}(\mathbf{x}, \mathbf{x}') + \gamma \mathcal{P}_\mathbf{x}(\mathbf{x})^\top \mathbf{w} - \mathbf{x}^\top \mathbf{w} \right), \tag{44}$$

where $\mathcal{P}_\mathbf{x} = \mathbb{E}\left[(\mathbf{x}')^\top|\mathbf{x}\right]$ is the forward expectation model and $\breve{r}_\mathbf{x}(\mathbf{x}, \mathbf{x}') = \mathbb{E}\left[\mathbf{x}'|\mathbf{x}\right] \Theta_r \mathbf{x}\mathbf{x}$ is the corresponding reward expectation model.

### C.2 Maximum Likelihood Estimation for Expectation Models

The expectation model approach trains a model predictor to output a point estimate as a function of the input. When trained with mean-squared error, the probabilistic interpretation is that the point estimate corresponds to the mean of a Gaussian distribution with fixed input-independent variance $\sigma^2$: $\mathcal{P}(\tilde{\mathbf{x}}|\mathbf{x}) = \mathcal{N}(\mu(\mathbf{x}); \sigma^2)$. Minimizing the negative log likelihood in this case leads to least-squares regression. Concretely, the empirical loss for the MLE objective of backward model learning is

$$l_{\text{MLE}}(\breve{\mathcal{P}}) = \frac{1}{n} \sum_{(\mathbf{x}_i, A_i, \mathbf{x}'_i) \in \mathcal{D}_n} \left| \breve{\mathcal{P}}(\mathbf{x}'_i) - \mathbf{x}_i \right|^2 \approx \mathbb{E}\left[ \left| \breve{\mathcal{P}}(\mathbf{x}') - \mathbf{x} \right|^2 \right], \tag{45}$$

where $\breve{\mathcal{P}} \leftarrow \text{argmin}_{\breve{\mathcal{P}}^\dagger \in \mathcal{P}} l_{\text{MLE}}(\breve{\mathcal{P}}^\dagger)$, $\mathcal{P}$ is the model space and $\mathcal{D}_n = \{(\mathbf{x}_i, A_i, R_i, \mathbf{x}'_i)\}_{i=1}^n$ represents collected data from interaction.

### C.3 Planner-aware Models

If the true model does not belong to the model estimator's space and approximation errors exist, a planner-aware method can choose a model with minimum error with respect to $\overline{\texttt{Planner}}$'s objective. Both forward and backward planning objectives for value-based methods try to find an approximation $v$ to $v_\pi$ by applying one step of semi-gradient model-based TD update. The planner-aware model-learning (PlanML) objective is less constrained than the MLE objective in that it only tries to ensure that replacing the true dynamics with the model

is inconsequential for the internal mechanism of $\overleftarrow{\texttt{Planner}}$. One choice of loss function of this objective could be:

$$l_{\text{PlanML}}(\breve{\mathcal{P}}) = \frac{1}{n} \sum_{(\mathbf{x}_i, A_i, \mathbf{x}'_i) \in \mathcal{D}_n} \left| \tilde{\Delta}^* - \tilde{\Delta} \right|^2 \approx \mathbb{E}\left[ \left| \nabla_{\mathbf{w}} \overleftarrow{\texttt{Planner}}(\breve{\mathcal{P}}^\star, v) - \nabla_{\mathbf{w}} \overleftarrow{\texttt{Planner}}(\breve{\mathcal{P}}, v) \right|^2 \right], \quad (46)$$

where the value updates in the empirical loss are: $\tilde{\Delta}^* = \left( R_i + \gamma v_{\mathbf{w}}\left(\mathbf{x}'_i\right) - v_{\mathbf{w}}\left(\mathbf{x}_i\right) \right) \nabla_{\mathbf{w}} v_{\mathbf{w}}\left(\mathbf{x}_i\right)$ – the true model update, and $\tilde{\Delta} = \left( \bar{r}\left(\breve{\mathcal{P}}\left(\mathbf{x}'_i\right), \mathbf{x}'_i\right) + \gamma v_{\mathbf{w}}\left(\mathbf{x}'_i\right) - v_{\mathbf{w}}\left(\breve{\mathcal{P}}\left(\mathbf{x}'_i\right)\right) \right) \nabla_{\mathbf{w}} v_{\mathbf{w}}\left(\breve{\mathcal{P}}\left(\mathbf{x}'_i\right)\right)$ – the update resulting from using the estimated internal model. While this requires the true value function $v$ in general, several approximations are possible, such as (a) using the current estimate of the value function resulting from pure planning $- v_{\mathbf{w}}^{\breve{\mathcal{P}}}$, or (b) using the current estimate value function resulting from direct learning and additional planning $- v_{\mathbf{w}}^{\breve{\mathcal{P}}+\breve{\mathcal{P}}^\star}$. Combining $\overleftarrow{\texttt{Planner}}$ with policy gradients (PG), similarly to (Abachi et al., 2020), might also be interesting.

Other objectives might be possible, such as directly parameterizing and estimating the expected parameter updates of the value function, thus learning a fully *abstract model*. Learning of this kind would shadow the internal arrow of time of the model. The ultimate unconstrained objective could meta-learn the model, such that, after a model learning update, the model would be useful for planning.

**Tabular planner-aware model learning**
We derive the gradient of a model which uses a planner-aware objective (similarly to Farahmand et al., 2017).The derivations hold for a probabilistic model and contrast the difference to a maximum likelihood model.

We consider the model to belong to an exponential family described by features $\tilde{\mathbf{x}} : \mathcal{S} \rightarrow \mathbb{R}^d$ and parameters $\theta$:

$$\breve{\mathcal{P}}_\theta\left(\tilde{s}|s,\right) = \frac{\exp\left(\tilde{\mathbf{x}}^\top\left(\tilde{s}, s\right)\theta\right)}{\sum_{\tilde{s}} \exp\left(\tilde{\mathbf{x}}^\top\left(\tilde{s}, s\right)\theta\right)} \quad (47)$$

Similarly, we take the value function to be a linear function of a (possibly) different set of features $\mathbf{x} : \mathcal{S} \rightarrow \mathbb{R}^d$ and parameters $\mathbf{w}$: $v_{\mathbf{w}}(s) = \mathbf{x}^\top(s)\mathbf{w}$. Then the PlanML loss is:

$$l_{\text{PlanML}}(\breve{\mathcal{P}}_\theta; s) = \left| \sum_{\tilde{s}} \left( \breve{\mathcal{P}}_\theta(\tilde{s}|s) - \breve{\mathcal{P}}^\star(\tilde{s}|s) \right) \left( r(\mathbf{x}(\tilde{s}), \mathbf{x}(s)) + \gamma \mathbf{x}(s)^\top \mathbf{w} - \mathbf{x}(\tilde{s})^\top \mathbf{w} \right) \mathbf{x}(\tilde{s})^\top \right|^2 \quad (48)$$

$$= \left| \sum_{\tilde{s}} \left( \breve{\mathcal{P}}_\theta(\tilde{s}|s) - \breve{\mathcal{P}}^\star(\tilde{s}|s) \right) \tilde{\Delta}(\tilde{s}, s) \right|^2 \quad (49)$$

Introducing an empirical measure of the state space corresponding to a distribution under which the data is sampled by the agent and the true environment transition dynamics $\breve{\mathcal{P}}^\star$ we have the observed empirical distribution:

$$l_{\text{PlanML}}(\breve{\mathcal{P}}_\theta) = \sum_{(\tilde{S}_i, \tilde{A}_i, R_i, S_i)} \left| \left( \mathbb{E}_{\tilde{S} \sim \breve{\mathcal{P}}_\theta(\cdot|S_i)}\left[\tilde{\Delta}(\tilde{S}, S_i)\right] - \tilde{\Delta}(\tilde{S}_i, S_i) \right) \right|^2 \quad (50)$$

The gradient of the estimated model's output with respect to its internal parameters is:

$$\nabla_\theta \breve{\mathcal{P}}_\theta(\tilde{s}|s) = \breve{\mathcal{P}}_\theta(\tilde{s}|s)\left[ \tilde{\mathbf{x}}^\top(\tilde{s}, s) - \sum_{\tilde{s}} \breve{\mathcal{P}}_\theta(\tilde{s}|s)\tilde{\mathbf{x}}^\top(\tilde{s}, s) \right] \quad (51)$$

As a result the gradient of the PlanML loss with respect to the model parameters is:

$$\nabla_\theta l_{\text{PlanML}}(\breve{\mathcal{P}}_\theta) = \frac{1}{n} \sum_{(\tilde{S}_i, \tilde{A}_i, R_i, S_i)} \left[ \mathbb{E}_{\tilde{S} \sim \breve{\mathcal{P}}_\theta(\cdot|S_i)}\left[\tilde{\Delta}(\tilde{S}, S_i)\right] - \tilde{\Delta}(\tilde{S}_i, S_i) \right] \left[ \sum_{\tilde{s}} \tilde{\Delta}(\tilde{s}, S_i)^\top \nabla_\theta \breve{\mathcal{P}}_\theta(\tilde{s}|S_i) \right]$$
$$(52)$$

$$= \frac{1}{n} \sum_{(\tilde{S}_i, \tilde{A}_i, R_i, S_i)} \left[ \mathbb{E}_{\tilde{S} \sim \breve{\mathcal{P}}_\theta(\cdot|S_i)}\left[\tilde{\Delta}(\tilde{S}, S_i)\right] - \tilde{\Delta}(\tilde{S}_i, S_i) \right] \quad (53)$$

$$\left[ \mathbb{E}_{\tilde{S} \sim \breve{\mathcal{P}}_\theta(\cdot|S_i)}\left[\tilde{\Delta}(\tilde{S}, S_i)^\top \tilde{\mathbf{x}}^\top(\tilde{S}, S_i)\right] - \mathbb{E}_{\tilde{S} \sim \breve{\mathcal{P}}_\theta(\cdot|S_i)}\left[\tilde{\Delta}(\tilde{S}, S_i)^\top\right] \mathbb{E}_{\tilde{S} \sim \breve{\mathcal{P}}_\theta(\cdot|S_i)}\left[\tilde{\mathbf{x}}^\top(\tilde{S}, S_i)\right] \right]$$
$$(54)$$

$$= \frac{1}{n} \sum_{(\tilde{S}_i, \tilde{A}_i, R_i, S_i)} \left[ \mathbb{E}_{\tilde{S} \sim \breve{\mathcal{P}}_\theta(\cdot|S_i)}\left[\tilde{\Delta}(\tilde{S}, S_i)\right] - \tilde{\Delta}(\tilde{S}_i, S_i) \right] \text{Cov}_{\tilde{S} \sim \breve{\mathcal{P}}_\theta(\cdot|S_i)}\left[\tilde{\Delta}(\tilde{S}, S_i)^\top \tilde{\mathbf{x}}^\top(\tilde{S}, S_i)\right],$$
$$(55)$$

where Cov can be expanded using the TD error $\delta$ as:

$$\text{Cov}_{\tilde{S}\sim\breve{\mathcal{P}}_\theta(\cdot|S_i)}\left[\breve{\Delta}(\tilde{S},S_i)^\top\tilde{\mathbf{x}}^\top(\tilde{S},S_i)\right] = \text{Cov}_{\tilde{S}\sim\breve{\mathcal{P}}_\theta(\cdot|S_i)}\left[\breve{\Delta}(\tilde{S},S_i)\mathbf{x}(\tilde{S},S_i)\tilde{\mathbf{x}}^\top(\tilde{S},S_i)\right] \tag{56}$$

$$= \mathbb{E}_{\tilde{S}\sim\breve{\mathcal{P}}_\theta(\cdot|S_i)}\left[\breve{\Delta}(\tilde{S},S_i)\mathbf{x}(\tilde{S}|S_i)\tilde{\mathbf{x}}^\top(\tilde{S},S_i)\right] - \mathbb{E}_{\tilde{S}\sim\breve{\mathcal{P}}_\theta(\cdot|S_i)}\left[\breve{\Delta}(\tilde{S},S_i)\mathbf{x}(\tilde{S},S_i)\right]\mathbb{E}_{\tilde{S}\sim\breve{\mathcal{P}}_\theta(\cdot|S_i)}\left[\tilde{\mathbf{x}}^\top(\tilde{S},S_i)\right] \tag{57}$$

In contrast the gradient of the MLE objective is:

$$\nabla_\theta l_{\text{MLE}}(\breve{\mathcal{P}}_\theta) = \frac{1}{n}\sum_{(\tilde{S}_i,\tilde{A}_i,R_i,S_i)}\left[\mathbb{E}_{\tilde{S}\sim\breve{\mathcal{P}}_\theta(\cdot|S_i)}\left[\tilde{\mathbf{x}}^\top(\tilde{S},S_i)\right] - \tilde{\mathbf{x}}^\top(\tilde{S}_i,S_i)\right] \tag{58}$$

For the forward case, a similar result can be derived:

$$l_{\text{PlanML}}(\mathcal{P}_\theta;s) = \left|\sum_{s'}\left(\mathcal{P}_\theta(s'|s) - \mathcal{P}^\star(s'|s)\right)\left(r(\mathbf{x}(s),\mathbf{x}(s')) + \gamma\mathbf{x}(s')^\top\mathbf{w} - \mathbf{x}(s)^\top\mathbf{w}\right)\mathbf{x}(s)^\top\right|^2 \tag{59}$$

$$= \left|\sum_{s'}\left(\mathcal{P}_\theta(s'|s) - \mathcal{P}^\star(s'|s)\right)\Delta(s,s')\right|^2 \tag{60}$$

Introducing an empirical measure of the state space corresponding to a distribution under which the data is sampled by the agent and the true environment transition dynamics $\mathcal{P}^\star$ with the observed empirical distribution:

$$l_{\text{PlanML}}(\mathcal{P}_\theta) = \sum_{(S_i,A_i,R_i,S'_i)}\left|\left(\mathbb{E}_{S'\sim\mathcal{P}_\theta(\cdot|S_i)}\Delta(S_i,S') - \Delta(S_i,S'_i)\right)\right|^2 \tag{61}$$

The gradient of the estimated model is the same as in the backward case and as a result the gradient of the forward PlanML loss is:

$$\nabla_\theta l_{\text{PlanML}}(\mathcal{P}_\theta) = \frac{1}{n}\sum_{(S_i,A_i,R_i,S'_i)}\left[\mathbb{E}_{S'\sim\mathcal{P}_\theta(\cdot|S_i)}\left[\Delta(S_i,s')\right] - \Delta(S_i,S'_i)\right]\left[\sum_{s'}\Delta(S_i,s')^\top\nabla_\theta\breve{\mathcal{P}}_\theta(s'|S_i)\right] \tag{62}$$

$$= \frac{1}{n}\sum_{(S_i,A_i,R_i,S'_i)}\left[\mathbb{E}_{S'\sim\breve{\mathcal{P}}_\theta(\cdot|S_i)}\left[\Delta(S_i,s')\right] - \Delta(S_i,S'_i)\right] \tag{63}$$

$$\left[\mathbb{E}_{S'\sim\mathcal{P}_\theta(\cdot|S_i)}\left[\Delta(S_i,S')^\top\tilde{\mathbf{x}}^\top(S',S_i)\right] - \mathbb{E}_{S'\sim\mathcal{P}_\theta(\cdot|S_i)}\left[\Delta(S_i,S')^\top\right]\mathbb{E}_{S'\sim\mathcal{P}_\theta(\cdot|S_i)}\left[\tilde{\mathbf{x}}^\top(S',S_i)\right]\right] \tag{64}$$

$$= \frac{1}{n}\sum_{(S_i,A_i,R_i,S'_i)}\left[\mathbb{E}_{S'\sim\mathcal{P}_\theta(\cdot|S_i)}\left[\Delta(S_i,S')\right] - \Delta(S_i,S'_i)\right]\text{Cov}_{S'\sim\mathcal{P}_\theta(\cdot|S_i)}\left[\Delta(S_i,S')^\top\tilde{\mathbf{x}}^\top(S',S_i)\right], \tag{65}$$

where Cov can be expanded as:

$$\text{Cov}_{S'\sim\mathcal{P}_\theta(\cdot|S_i)}\left[\Delta(S_i,S')^\top\tilde{\mathbf{x}}^\top(S',S_i)\right] = \text{Cov}_{S'\sim\mathcal{P}_\theta(\cdot|S_i)}\left[\delta(S_i,S')\mathbf{x}(S_i)\tilde{\mathbf{x}}^\top(S',S_i)\right] \tag{66}$$

$$= \mathbb{E}_{S'\sim\mathcal{P}_\theta(\cdot|S_i)}\left[\delta(S_i,S')\mathbf{x}(S_i)\tilde{\mathbf{x}}^\top(S',S_i)\right] - \mathbb{E}_{S'\sim\mathcal{P}_\theta(\cdot|S_i)}\left[\delta(S_i,S')\mathbf{x}(S_i)\right]\mathbb{E}_{S'\sim\mathcal{P}_\theta(\cdot|S_i)}\left[\tilde{\mathbf{x}}^\top(S',S_i)\right] \tag{67}$$

In contrast the gradient of the forward MLE objective is:

$$\nabla_\theta l_{\text{MLE}}(\mathcal{P}_\theta) = \frac{1}{n}\sum_{(S_i,A_i,R_i,S'_i)}\left[\mathbb{E}_{S'\sim\mathcal{P}_\theta(\cdot|S_i)}\left[\tilde{\mathbf{x}}^\top(S',S_i)\right] - \tilde{\mathbf{x}}^\top(S'_i,S_i)\right] \tag{68}$$

# D Empirical Studies Details

The first setting in which we illustrate the antithesis between forward and backward planning is a Random Markov Chain consisting of a leveled state space with one (or more) sets of states (or levels): $\{x_i\}_{i \in [0:n_x]}$ and $\{y_j\}_{j \in [0:n_y]}$ (Fig. 2), where we vary $n_x$ and $n_y$ in our experiments . We additionally experiment with adding intermediary levels: $\{z_k^l\}_{k \in [0:n_z^l], l \in [1:L]}$, where $L$ is the number of levels and $n_z^l$ is the size of level $l$. The states from a particular level transition only to states in the next level, thus establishing a particular flow and stationary structure of the Markov Chain under study. The transition probabilities between the leveled sets of states are sampled randomly from a uniform distribution $\mathcal{U}(0,1)$ and normalized accordingly. The states $\{y_j\}_{j \in [0:n_y]}$ are terminal; their rewards are dependent on both ends of the transition and are sampled from a normal distribution $\mathcal{N}(10,10)$. The criteria we use to evaluate the quality of the estimated models is the root mean squared value error (RMSVE): $\sqrt{(|v_\pi - v|_2^2)}$, which shows how close $v_{\mathcal{P}}$ and $v_{\bar{\mathcal{P}}}$ are to $v_\pi$. The experiments are ablation studies of the effects of varying the *fan-in* ($n_x$), the *fan-out* ($n_y$) and the number of levels $l$ with their corresponding sizes $n_z^l$.

For this investigation we performed two types of experiments: (i) on 3-level bipartite graphs as illustrated in figure 1-Left,Center-Left (ii) on 2-level bipartite graphs as shown in figure 1-Center-Right, Right. All experiments start with learning rates of 1.0 for model-free learning, planning and model-learning, which are linearly decayed over the course of learning. For (a) the channelling structure we use is: $L = 1, n_x = 500, n_z^1 = 50, n_y = 5$, whereas the broadcasting pattern has the opposite attributes: $L = 1, n_x = 5, n_z^1 = 50, n_y = 500$. For (b) the results reported are for $(n_x, n_y) \in \{(500,5),(50,5),(5,5),(5,50),(5,500)\}$, labeling the x-axis with the simplified ratio. Results shown in the main text are averaged over 20 seeds and show the standard error over runs.

---

**Algorithm 4:** Online Backward-Dyna: Learning & Backward Planning

---
1: **Input** policy $\pi, n$
2: $s \sim \text{env}()$
3: **for** each interaction $\{1, 2 \ldots T\}$ **do**
4:     $a \sim \pi(x)$
5:     $r, \gamma, s' \sim \text{env}(a)$
6:     $\bar{\mathcal{P}}, \bar{r} \leftarrow \text{model\_learning\_update}(s, a, s')$
7:     $v \leftarrow \text{learning\_update}(s, a, r, \gamma, s')$
8:     $s_{\text{ref}} \leftarrow \text{planning\_reference\_state}(s, s')$
9:     **for** each $\tilde{s} \in \mathcal{S}$ **do**
10:         $y = \bar{r}(\tilde{s}, s_{\text{ref}}) + \gamma v(s_{\text{ref}})$
11:         $\bar{\Delta}(\tilde{s}) = \bar{\mathcal{P}}(\tilde{s}|s_{\text{ref}})\,(y - v(\tilde{s}))$
12:         $v(\tilde{s}) \leftarrow v(\tilde{s}) + \alpha \bar{\Delta}(\tilde{s})$
13:     $s \leftarrow s'$

---

All experiments use, for any transition $s \xrightarrow{a} s'$, the following reference frame for planning: background models – the current state $s'$ of a transition, forward models – the previous state $s$ of a transition. The exact definition of reference frames is given in the main text and below, in the next experiment. To make things more concrete, we include the pseudo-code for the backward planning algorithm used for this experiment in Algorithm 4 (forward planning is done similarly, see Alg. 2 and 3 for more details on the differences).

## D.1 Indirect effect of backward planning on control

In the following experiments, we perform ablation studies on the discrete navigation task from (Sutton and Barto, 2018) illustrated in Fig. 5. "G" marks the position of the goal and the end of an episode. "S" denotes the starting state to which the agent is reset at the end of the episode. The maximum size of episodes is set to 400 to allow the agent sufficient exploration. The state space size is 48, $\gamma = 0.99$. There are 4 actions that can transition the agent to each one of the adjacent states.

Figure 5: **Maze Navigation**: Illustration of the MDP used in the control experiments.

The following two experiments use algorithms that perform *background planning* with forward and backward mechanisms. Specifically, for the forward case we use algorithm 2 and for the backward case algorithm 3. The control algorithms interlace model-free learning of the optimal action-value function $q$ (line 7 – forward & backward) with model-learning (of forward and backward models respectively – line 6) and additional planning steps (forward and backward – lines 8:10). The model-free ''learning_update'' performs q-learning updates of the form:

$$q(s,a) = q(s,a) + \alpha \left( r\left(s, a, s'\right) + \gamma \max_{a'} q\left(s', a'\right) - q\left(s, a\right) \right). \tag{69}$$

The models are estimated using MLE – for the transition dynamics (and for the termination function $\bar{\gamma}$ in the case of forward models) and regression - for the reward models. The planning processes add model-based forward and backward updates using the estimated models. Forward planning updates perform an expected

planning-update using the forward models $(\mathcal{P}, \bar{r}, \bar{\gamma})$, $\forall s \in \mathcal{S}, \forall a \in \mathcal{A}$:

$$q(s_{\text{ref}}, a) = q(s_{\text{ref}}, a) + \alpha \left( \sum_{s' \in \mathcal{S}} \mathcal{P}(s'|s_{\text{ref}}, a) \left( \bar{r}(s') + \bar{\gamma}(s') \max_{a'} q(s', a') \right) - q(s_{\text{ref}}, a) \right) \qquad (70)$$

and the backward planning process adds the following updates according to the backward models $(\breve{\mathcal{P}}, \bar{r}, \forall \tilde{s} \in \mathcal{S}, \forall a \in \mathcal{A}$:

$$q(\tilde{s}, \tilde{a}) = q(\tilde{s}, \tilde{a}) + \alpha \breve{\mathcal{P}}(\tilde{s}, \tilde{a}|s_{\text{ref}}) \left( \overleftarrow{r}(s_{\text{ref}}) + \gamma \max_{\bar{a}} q(s_{\text{ref}}, \bar{a}) - q(\tilde{s}, \tilde{a}) \right). \qquad (71)$$

The procedure specified in the pseudocode with ``planning_reference_state'' is the subject of the first experiment we performed for this control setting.

**Planning frame of reference**
We first ask ourselves whether the frame of reference from which the agent starts planning matters. When it encounters a transition $s \xrightarrow{a} s'$, it could plan from either $s$ or $s'$. This choice represents the output of the procedure ``planning_reference_state'' in the pseudocode. In our experiments, we perform ablation studies for pure planning, i.e. without the additional model-free procedure ``learning_update'', and for the full learning framework described in the pseudocode. The experiments are performed with deterministic dynamics. The learning rates used for q-learning, model learning and planning are started at 1.0 and linearly decayed over the course of training. The policy used for acting is $\epsilon$-greedy with $\epsilon$ decayed linearly over the course of training from 0.5 to 0. Results shown in the main text are averaged over 20 seeds and show the standard error over runs.

**Changing the level of stochasticity**
We next investigate how stochasticity affects performance. The four settings we investigate are : (a) deterministic dynamics – when the environment transitions the agent to the intended direction with probability 1; similarly reward is +1 with probability 1; (b) stochastic dynamics – reward is still deterministic, yet the environment transitions the agent to a random adjacent state with probability 0.5; stochastic rewards – transition dynamics are deterministic, yet the rewards are +1 with probability 0.5, otherwise 0; (c) stochastic rewards identical to the previous setting; rewards are +1 with probability 0.1. We apply the same algorithms described in the previous section taking $s$ to be the reference frame for backward planning and $s'$ for forward planning. Table 1 specifies the learning rates used for q-learning, model learning and planning for each of the four settings we described above. These are linearly decayed over the course of training. The policy used for acting is $\epsilon$-greedy with $\epsilon$ decayed linearly over the course of training from 0.5 to 0. Results shown in the main text are averaged over 20 seeds and show the standard error over runs. Hyperparameters have been chosen from $\{1.0, 0.5, 0.1, 0.05, 0.01\}$.

| **Environment Setting** | $\alpha$**(learning&planning)** | $\alpha_m$**(model learning)** |
|---|---|---|
| Deterministic rewards and transitions | 1.0 | 1.0 |
| Deterministic transitions/stochastic rewards (p=0.5) | 0.1 | 0.5 |
| Deterministic transitions/stochastic reward (p=0.1) | 0.05 | 0.05 |
| Stochastic transitions (p=0.5)/deterministic rewards | 0.1 | 0.5 |

Table 1: Hyperparameters for tabular control on the Maze Gridworld

## Footnotes

[3] Learning the $\lambda$ function is part of the *'discovery problem'* and the best strategy still unclear, although a meta-learning strategy is perhaps effective (Xu et al., 2018).