[Reviews · NeurIPS 2020]

Review 1

Summary and Contributions: The paper proposes backward model based reinforcement learning (RL) algorithms. They show that backward model based algorithms are equivalent to corresponding forward models in terms of state transitions and value functions. With empirical studies, the paper shows that backward models are beneficial when many different states lead to a common state (large fan-in, small fan-out).

Strengths: - The paper proposes an interesting direction of algorithm design with backward models. - All the claims in the paper is backed by mathematical derivations in Appendix. - Empirical experiments are designed for comparative analysis between the classic algorithms and proposed methods

Weaknesses: The paper could benefit from clear examples to explain the concepts introduced, and a better differentiation from some of the closely related works.

Correctness: Yes

Clarity: The paper would benefit from an example to explain the concepts introduced. Instead of comparing against prior work throughout the paper, it will be easier to read if the paper explains the proposed concepts clearly. The related work section can be used for differentiation and comparisons. It is not clear what an abstract model is.

Relation to Prior Work: The related work section is mostly a list of papers. There is not much context provided, and the authors do not attempt to differentiate from the prior works. However, adequate context has been provided throughout the paper. However, certain closely related works such as Goyal et al., Satija et al, is not discussed and differentiated against.

Reproducibility: Yes

Additional Feedback: The authors reference high dimensional problems several times. It will be interesting to see empirical results in standard benchmarks. Post Rebuttal response: The author response has adequately addressed my concerns. I hope they revise the related work section with explanation of how their work distinguishes from closely related works as explained in the rebuttal.


Review 2

Summary and Contributions: This paper is a study on the use of forward and backward planning and their trade-offs as used in model-based reinforcement learning. The authors particularly highlight the properties of the somewhat less appreciated hindsight planning. They show that forward and backward planning are in some sense complementary depending on the environment’s dynamics, e.g. fan-in/fan-out MDP structure. They highlight various ways of integrating planning with model-free updates and their differences in stochastic environments. All this is shown in two illustrative toy experiments.

Strengths: This work is a very thorough and in-depth study of an important problem in model-based reinforcement learning. The authors introduce an extensive formalism to frame the general problem and embed their work in an in-depth coverage of related work. I have not seen such a detailed comparison of forward versus backward planning as shown here. This work is both valuable as a novel contribution and as a review of the studied problem. The empirical rigor is commendable, the authors provide detailed description of all experiments as well as their code. For all results, they run experiments with many different seeds. Given the low computational cost, this is quite plausible to do, and very much appreciated.

Weaknesses: I have found the manuscript to be very dense and sometimes hard to read, in particular Section 3 model estimation and the empirical section can only be appreciated together with Appendix. For its content, this section is very short, and somewhat lacks in giving intuition and explanation, focusing mostly on claims and reporting results. The studied problem focus entirely on simple problems, an inclusion of a more difficult problem (e.g. continuous) showcasing one or two of the findings would have been a great addition and would convince the reader these findings apply more generally. However, I can see that performance here may be dependent on a lot of confounding factors which would make it harder to analyse.

Correctness: I have found no technical error. Empirical rigor seems to be very high.

Clarity: At times, the paper seems overly technical, taking a lot of time to state something well known in a long (but correct) fashion. This is a stylistic choice, of course, at parts I could have lived with a shorter statement or omission in Section 2 and 3 for a bit more in-depth discussion of the findings in Section 4. Section 3, the subsection about model estimation left me a bit confused at various points. It lists a lot of various possible options one has for various components, but I'm unsure to what end they are presented. They are not further studied in the experimental section. Some small errors: 190: constrains => constraints 277: is => it 755: hyphen/dash length 757: bellow => below

Relation to Prior Work: The authors performed an extensive literature review of both the state of the art and historical development of the studied problem. The authors cite van Hasselt et al “When to use parametric models in reinforcement learning?” as both a motivation for their work and experiments. I’d have liked some discussion or comparison of the results in this context. In particular 4 (iii) seem to have qualitatively identical results to this prior work at least with relation of forward and backward planning. Model-free results, however, seem a bit different.

Reproducibility: Yes

Additional Feedback: In Figure 3, deterministic environment: bw_plan seems to converge slightly faster than fw_plan even when given the correct model. This seems to go against your conclusion that forward planning is invaluable when the future is predictable (i.e. model is good and the environment deterministic). When trying to run the provided code, additionally to the packages specified in README.md, one has to install absl-py, jaxlib, tqdm and dm-haiku. The codebase folder has to be added to the python path. Then, I run into various import errors of, first of bw_PAML_qt then VanillaQ. Here, I stopped trying. I've read the claim made in 128-129 of backward models updating fake states being less harmful numerous times, but I don't think I have seen much evidence for this. It makes intuitive sense, of course, to some extent. However, especially when powerful function approximators are used, they have a tendency to map any input onto the same manifold. Your experimental results in stochastic settings supports that there is something better about backward planning but have you actually looked at the state distribution of the resulting updates? Are they actually fake/unused states? And, have you observed the same when using e.g. a deeper neural network or a sequence model?


Review 3

Summary and Contributions: This paper looks at forward and backward dynamics models for the benefit of "Dyna-style" reinforcement learning, and interestingly show in which cases either model is more useful in.

Strengths: The strength of this work is the initial explanations of the backwards model, which were clear and interesting. Also the experiment ablations were excellent, with a lot of the questions one would expect to ask.

Weaknesses: The main weakness I found was lack of experiential setup explanations. I had trouble understanding the experiments, namely what "fan-in, fan-out" even was, and what was the task? I'm happy to update my scores based on changes to this and look forward to the authors' replies.

Correctness: Correct (as far as I know)

Clarity: Clear writing, but details missing.

Relation to Prior Work: Prior work was clearly discussed.

Reproducibility: Yes

Additional Feedback: major points: lines 234-246: what is "fan in" and "fan out", this paragraph doesn't explain what it is? Does it relate to neural network architecture (shown in Figure 1)? I have no idea what "large fan-in" and "small fan-out" means. Is channeling a bottleneck in the state space? If so, where is this? For the initial experiments, what environment was being used? medium points: lines 127-129: I still do not understand why the backwards model can "sidestep" the issues of lines 124-126. To damage a "fake" state means, it's (1) giving incorrect estimates for another state that we may need to use later, and (2) not providing an update for the correct previous state? lines 140-141: "tethered to a policy": true if the idea is to predict the action taken, but not if it were p(x_t | a_t, x_{t+1}), which can also be called a "backwards model". Possibly worth clarifying earlier what is meant by "backwards model" in this case? lines 143-145: that is surprising! line 153: perhaps mention this earlier too? small points: Equation 2: state "using Bayes rule" to get this? Figure 1, left-side legend: how come "fw_plan" has a "bw" / backward arrow over P? And is "bw_plan" missing a backward arrow over P, which the right-side legend has? ====== After author response ====== "on explanatory details": I thank the authors for the additional explanation, and encourage the authors to add into the camera-ready copy. This addresses one of my concerns, so I'll update my score 5-->6. My remaining concern is below. "backward planning being less harmful": I agree this makes sense in the case of "unreachable" states, but what guarantees all the "backwards mistakes" will always be to "unreachable" states? (I assume you mean "always unreachable from any state", not just the state in question. But if not, then my critique still holds). The general case is the mistakes would potentially be to reachable states, no? I appreciate your points about DeepRL not necessarily being required, but I do wonder if this issue of "[un]reachability" here arises more in continuous states spaces, which the grid-world experiments were able to avoid dealing with.


Review 4

Summary and Contributions: The paper proposes a backward planning model for hindsight credit assignment and analyzed the model on synthetic tasks.

Strengths: 1. The paper is well written and easy to follow. 2. It addresses an interesting problem in RL (hindsight credit assignment). Instead of performing credit assignment on sampled predecessor states, the paper proposes to build a backward generative (transition) model that can assign credit to a distribution of states. 3. The experiments on the synthetic tasks are interesting.

Weaknesses: 1. I like the approach of the paper and it addresses an important problem. However, although experiments on synthetic datasets help to understand the behaviour of the proposed model, I would still like to see some experiments on real RL tasks. The current experiments seem to be fairly limited. 2. There seem to be some missing citations/ related works that should be discussed. For example, SAB [1]. proposes a sparse attention model with skip connections through time for hindsight credit assignment. Temporal value transport [2] that performs credit assignment through time (with skip connections). [1]. Ke, Nan Rosemary, et al. "Sparse attentive backtracking: Temporal credit assignment through reminding." Advances in neural information processing systems. 2018. [2]. Hung, Chia-Chun, et al. "Optimizing agent behavior over long time scales by transporting value." Nature communications 10.1 (2019): 1-12.

Correctness: The method seems to be correct.

Clarity: The paper is very well written and easy to follow.

Relation to Prior Work: Some prior works are missing, as stated earlier. [1]. Ke, Nan Rosemary, et al. "Sparse attentive backtracking: Temporal credit assignment through reminding." Advances in neural information processing systems. 2018. [2]. Hung, Chia-Chun, et al. "Optimizing agent behavior over long time scales by transporting value." Nature communications 10.1 (2019): 1-12.

Reproducibility: Yes

Additional Feedback:

[Author Response · NeurIPS 2020]

We thank all of the reviewers for their useful and insightful feedback. We are encouraged they found our work interesting (R1, R2, R3, R4), novel (R2) and scientifically rigorous (R2, R3). We are happy all reviewers recognize the importance of the problem we address. We are glad they found our work impactful in the design of new algorithms (R1), that our empirical studies were convincing (R1, R2, R3) and that they proactively answer their questions (R3). We address reviewer comments below and will incorporate all feedback.

@R1*"clear examples to explain the concepts"*: Our work stands on the intuition that an agent might benefit from using its computation budget to spread information as fast as possible through the state space to all precursors of the current state, rather than just the one state and action that brought it to the current state on this occasion. An **analogy to human memory consolidation** may help: we are arguably good at retrospectively inferring (or inventing) the underlying causes of our experience and ruminating over them so as to improve our future predictions; ***we hypothesize agents could also use retrospective knowledge about the world (backward models) to consolidate (hindsight planning) their prospective knowledge (value functions)***. We will complement the **thought experiment** in Section 3 with this. Thank you for the suggestion. @R1*"not clear what an abstract model is"*: we define a model as being abstract when we remove inductive biases (e.g. structural constraints) in its construction and use, leaving it as a (learnable) black box. We will expand in the paper.

@R1,R2 _on related work_: The closest work to ours is that of **van Hasselt et. al.**, our results complement theirs: in the control experiment we extend by showing backward planning is *more robust in dealing with rare events (e.g. stochastic rewards) and different levels of stochasticity in the transitions*;@R2 van Hasselt et. al. compare against replay – a non-parametric model-based approach, NOT against model-free learning. **Goyal et. al.** use imitation learning on a generative model's outputs so as to improve exploration by incentivizing the agent towards the high value states on which the model was trained; in contrast, *we aim to formalize and tease apart the fundamental properties of **online** hindsight planning*. **Satija et. al.** use backward value functions to encode constraints for solving constrained MDPs (CMDPs) with safe policy improvements; though both our work and theirs employ some form of retrospective knowledge, *the contents, purposes and uses differ*. @R4 we will add the _missing citations_ and discuss how they relate to our work. Thank you for spotting them.

@R3, R2 _on explanatory details_: **Details on experimental settings are in appendix D**. The prediction experiment is run on a leveled state space s.t. transition dynamics between states generate bipartite graphs; we vary the no. of states on each level and the no. of levels to generate different structural properties. We refer to **fan-in/fan-out** as *the no. of predecessors/successors a state might have in the state space*. Thumbnails depict the phenomena of transitioning from a larger no. of predecessors that "funnel" into a smaller no. of successors, and vice-versa. **Task** $\equiv$ *value prediction*. We will make the main paper more clear by adding more details from the appendix. Thank you for highlighting this.

@R2 _utility of forward planning when the future is predictable_: The claim rests on the prediction experiment; the deterministic control setting is more conflated, performance difference too marginal, task too easy and backward models are also in their best regime. @R2 _codebase_: Missing requirements can be installed from pip and the *import bw_PAML_qt* safely deleted. @R2, R3 _backward planning being less harmful_: Great question! An erroneous forward model that predicts an unreachable state will move the value of a real state towards the arbitrary value of the bootstrapped state; an erroneous backward model will harmlessly distribute the value of a bootstrapped real state to an unreachable arbitrary-value state (see concurrent work Jafferjee et. al. - *"Hallucinating value..."*). @R3*"tethered to a policy"*: Both $\overleftarrow{p}_\pi(x_t, a_t | x_{t+1})$ and $\overleftarrow{p}_\pi(x_t | a_t, x_{t+1})$ are backward models. Eq (3) shows how the latter still **depends on $\pi$ through the stationary distributions** $\eta_\pi$. @R3*"using Bayes rule"* **Yes** (see appendix A). @R1,R2,R3 we appreciate pointing out the misplaced arrow in Fig.1's legend, detailed feedback on typos and suggestions on how to improve clarity.

@R1,R2,R3,R4*"deep RL experiments"*: Deep RL comes with **confounding factors**, e.g. the environment-dependent mixing time characterizing the correlated datastream forces the use of reply buffers to decorrelate experience and target networks to stabilize learning for incremental-update algorithms; this in turn pushes learning in the off-policy regime where convergence is only serendipitous; common testbeds are not informative of how different components of algorithms influence learning. We have left as future work extensions to function approximation and more complex problems, as we reckon (substantial) additional research is required in terms of testbeds and ablations to allow for **scientifically relevant hypotheses**. Backward models came with their own peculiarities and available choices in terms of estimation and use. Understanding and formalizing them was, in our view, **the first step in the right direction**, i.e. transferring to more complex problems in a principled way. **Part of our scientific contribution** was this analysis, revealing many options w.r.t model-learning objectives and planning-strategies (Section 3). The **goal** of our empirical studies was to disentangle core properties of these approaches so as to inform on the design of complex testbeds, where we can understand planning methods. We strongly believe that our contributions **set the stage for principled deep RL investigations**.

We can clarify these points in the paper and strengthen the discussion of existing literature. We feel the technical contribution is sound, surprising (to us at least) and potentially impactful to both scientific understanding and practice.

[Meta-Review · NeurIPS 2020]

Summary: The paper proposes to use backward planning for credit assignment in RL. Given an observed state, its Q-value is back-propagated to all potential past states from which the observed state could have resulted. A backward probability distribution of all past states that end up in a given state is used for this purpose. The paper also compares this backward strategy to the classical forward planning, and experiments on toy examples demonstrate the advantages of this method. Pros: - The idea of backward planning for credit assignment is interesting - Rigorous derivation of the algorithm - Empirical evaluation shows the the advantage of the method Cons: - Toy experiments - Lack of examples to explain the method Discussion and decision: The reviewers agree that this is a good paper. The idea is intriguing. One issue raised by the reviewers is the toyish nature of the problems considered in the experiments. The MDPs considered in the experiments have a very small number of states. The proposed algorithm needs to be tested on more serious problems and more realistic systems, similar to what is being considered in state-the-art works on RL, especially given that the proposed approach has no theoretical guarantees to it.